# Biochemical and functional characterization of mutant KRAS epitopes validates this oncoprotein for immunological targeting

Adham S. Bear [1,2✉], Tatiana Blanchard[3], Joseph Cesare [4], Michael J. Ford[5], Lee P. Richman[2], Chong Xu[3], Miren L. Baroja[3], Sarah McCuaig [6], Christina Costeas[6], Khatuna Gabunia[3], John Scholler[3], Avery D. Posey Jr. [3,7,8], Mark H. O'Hara[1,2], Anze Smole[3], Daniel J. Powell Jr. [3,9], Benjamin A. Garcia[10], Robert H. Vonderheide[1,2,3,11,12], Gerald P. Linette [1,3,11,12] & Beatriz M. Carreno [3,9,11,12✉]

Activating RAS missense mutations are among the most prevalent genomic alterations observed in human cancers and drive oncogenesis in the three most lethal tumor types. Emerging evidence suggests mutant KRAS (mKRAS) may be targeted immunologically, but mKRAS epitopes remain poorly defined. Here we employ a multi-omics approach to characterize HLA class I-restricted mKRAS epitopes. We provide proteomic evidence of mKRAS epitope processing and presentation by high prevalence HLA class I alleles. Select epitopes are immunogenic enabling mKRAS-specific TCRαβ isolation. TCR transfer to primary CD8$^+$ T cells confers cytotoxicity against mKRAS tumor cell lines independent of histologic origin, and the kinetics of lytic activity correlates with mKRAS peptide-HLA class I complex abundance. Adoptive transfer of mKRAS-TCR engineered CD8$^+$ T cells leads to tumor eradication in a xenograft model of metastatic lung cancer. This study validates mKRAS peptides as bona fide epitopes facilitating the development of immune therapies targeting this oncoprotein.

[1] Division of Hematology-Oncology, Department of Medicine, Perelman School of Medicine, University of Pennsylvania, Philadelphia, PA, USA. [2] Abramson Cancer Center, University of Pennsylvania, Philadelphia, PA, USA. [3] Center for Cellular Immunotherapies, Perelman School of Medicine, University of Pennsylvania, Philadelphia, PA, USA. [4] Department of Biochemistry and Biophysics, University of Pennsylvania, Philadelphia, PA, USA. [5] MSBioworks, Ann Arbor, MI, USA. [6] Perelman School of Medicine, University of Pennsylvania, Philadelphia, PA, USA. [7] Department of Systems Pharmacology and Translational Therapeutics, Perelman School of Medicine, University of Pennsylvania, Philadelphia, PA, USA. [8] Corporal Michael J. Crescenz VA Medical Center, Philadelphia, PA, USA. [9] Department of Pathology and Laboratory Medicine, Perelman School of Medicine, University of Pennsylvania, Philadelphia, PA, USA. [10] Department of Biochemistry and Biophysics, Epigenetics Institute, University of Pennsylvania, Philadelphia, PA, USA. [11] Parker Institute for Cancer Immunotherapy, Perelman School of Medicine, University of Pennsylvania, Philadelphia, PA, USA. [12]These authors jointly supervised this work: Robert H. Vonderheide, Gerald P. Linette, Beatriz M. Carreno. ✉email: adham.bear@pennmedicine.upenn.edu; bcarreno@upenn.edu

The RAS family of GTPases (KRAS, NRAS, HRAS) is mutated in approximately 20% of all human malignancies[1,2]. The vast majority of RAS genomic alterations are the result of missense mutations at codon positions G12, G13, or Q61 that involve the RAS protein GTP-binding domain, resulting in the activation of downstream effector substrates ERK and PI3-K leading to dysregulated cell growth and survival[3]. Among cancers in which KRAS mutations predominate, including pancreatic ductal adenocarcinomas (PDA), lung adenocarcinomas (LAC), and colorectal carcinomas (CRC), over 75% of amino acid substitutions occur at the G12 codon position[4]. The high frequency of G12 amino acid substitutions makes this codon position an ideal drug target; however, no small molecule inhibitors of G12 variants have been successfully developed aside from KRAS G12C that has only recently demonstrated clinical promise[5,6].

Somatic gene mutations within cancer cells may be translated into peptides that are processed and presented on the surface of tumor cells[7,8]. These mutated peptides can serve as foreign epitopes, or neoantigens, that may be recognized by αβ T cells of the host immune system. Neoantigen-specific T cell responses have been well documented in patients for whom immune checkpoint blockade therapy has been successful, and they are believed to be the key mediators of anti-tumor activity[9]. Neoantigen-specific T cells can be isolated from the peripheral blood or tumor tissue of antigen-exposed cancer patients[10], induced and expanded from the peripheral blood of cancer patients following neoantigen vaccination[11], and generated in vitro utilizing the naive T-cell receptor (TCR) repertoire of healthy donors[12]. These observations have garnered interest in the development of neoantigen-targeted cancer vaccines and adoptive T cell therapies.

Neoantigen-based treatment strategies are inherently highly personalized as somatic tumor mutations are rarely shared between patients[13,14]. The high prevalence and conserved mutational profile of KRAS affords a unique opportunity to develop a neoantigen-targeted therapy with broad generalizability. Mutant KRAS (mKRAS) has been previously studied as a potential target of cancer vaccines, and clinical studies have demonstrated the successful generation of CD4[+] and CD8[+] αβ T cell responses with reactivity against allogeneic or autologous mKRAS tumor cell lines[15–18]. More recently, mKRAS-specific T cells have been isolated and characterized from the peripheral blood of patients with mKRAS epithelial cancers[19,20], and they have been induced in vaccinated HLA-transgenic mice[21]. The therapeutic potential of targeting mKRAS as a cancer neoantigen was highlighted in a case report demonstrating clinical benefit in a patient with KRAS G12D metastatic CRC following the adoptive transfer of KRAS G12D-specific T cells restricted to HLA-C*08:02[22]. However, mKRAS as an immunological target remains poorly characterized, and there is a dearth of evidence regarding tumor cell processing and presentation of mKRAS-derived epitopes to guide the development of targeted immunotherapies.

In this work, we aim to validate mKRAS as an immunological target. Utilizing computational epitope prediction, biochemical assays, and proteomic analysis, we predict and identify high-affinity and/or high-stability mKRAS G12 peptides to HLA-A*03:01, HLA-A*11:01, and HLA-B*07:02, and we confirm these epitopes as constituents of their corresponding HLA class I ligandome. The induction of mKRAS-reactive T cells from healthy donors by select peptides both confirms peptide immunogenicity and enables the isolation of mKRAS-specific TCRs directed against the following three epitopes: G12V/HLA-A*03:01, G12V/HLA-A*11:01, and G12R/HLA-B*07:02. Expression of these mKRAS-specific TCRαβ pairs in a NFAT-inducible reporter T cell line or TCRαβ[null] primary CD8[+] T cells serves as sensitive probes to characterize the presentation of mKRAS G12 epitopes by tumor cell lines. TCR-redirected T cells display cytotoxic function against mKRAS tumor cell lines of various histologies without reactivity to wild-type (WT) KRAS peptides, thereby authenticating mKRAS G12V and G12R peptide ligands as bona fide neoantigens. Finally, the adoptive transfer of TCR redirected CD8[+] human T cells specific for G12V/HLA-A*03:01 or G12V/HLA-A*11:01 leads to tumor eradication and prolongs survival in a mouse xenograft model of metastatic lung cancer. These findings provide further validation and strengthen the nomination of mKRAS as an immunological target.

## Results

**Identification and characterization of mKRAS neoepitopes.** Although predictive peptide-HLA (p-HLA) binding affinity and stability are important criteria for neoantigen nomination[23–25], only a small percentage of the human proteome is presented by HLA molecules[26]. Therefore, only a fraction of candidate neoantigens are constituents of the tumor HLA ligandome[27], and predicting high-affinity/stability p-HLA complexes is not indicative of intracellular processing and presentation[28]. To improve the identification of bona fide mKRAS neoepitopes, we employed a comprehensive multi-omic approach consisting of (1) computational epitope prediction, (2) biochemical assessment of p-HLA binding / stability, (3) proteomic validation of antigen processing and presentation, and (4) characterization of immunogenicity (Supplementary Fig. 1).

We utilized the neoantigen prediction tool antigen.garnish[24] to perform a comprehensive identification of candidate HLA class I allele-restricted mKRAS G12 epitopes with recurrent amino acid substitutions (G12C, G12D, G12R, G12V) embedded within the 24 amino acid sequence derived from the WT KRAS protein. Epitopes derived from these 24-mer amino acid sequences were evaluated for predicted binding to HLA class I alleles to nominate HLA-A, HLA-B, and HLA-C nonamer (9-mer) and/or decamer (10-mer) mKRAS peptides (Supplementary Data 1, candidates <500 nM affinity). Predicted epitopes restricted to HLA-A*02:01, HLA-A*03:01, HLA-A*11:01, and HLA-B*07:02 were selected for further study given the high prevalence of these HLA class I alleles within the USA population (Fig. 1a). Peptides encoding these candidate epitopes are designated by their KRAS protein amino acid positions followed by the G12 amino acid substitution (Fig. 1b). The predicted HLA class I affinities and p-HLA complex stabilities of candidate epitopes are presented in Supplementary Tables 1 and 2.

We experimentally assessed p-HLA binding affinity and complex stability of candidate peptides as these two properties improve the performance of T cell epitope identification[29]. Furthermore, p-HLA complex stability positively correlates with immunogenicity[12]. To evaluate p-HLA binding affinity, we employed competitive peptide-binding assays using a FITC-labeled reference peptide and purified soluble HLA class I molecules to quantitate the binding capacity of non-labeled mKRAS peptide candidates. Peptide affinities are reported as concentrations of unlabeled peptide that result in 50% binding inhibition of FITC-labeled reference peptide (Supplementary Fig. 2 and Supplementary Table 1)[30]. To evaluate p-HLA complex stability, dissociation assays were performed where the stability of p-HLA complexes is expressed as the half-life of beta-2-microglobulin dissociation from the HLA heavy chain (Supplementary Table 2)[29]. Both assays incorporated validated HLA class I-restricted T cell epitopes as positive controls (Supplementary Table 3). Affinity binding values determined in competitive assays are reported as $\log_{10}[IC_{50}(nM)]$ and $IC_{50}(nM)$ and are defined according to a previously reported scale detailed in Supplementary Tables 1 and 3. For HLA-A*02:01 epitopes, only peptide 5-14V exhibited strong class

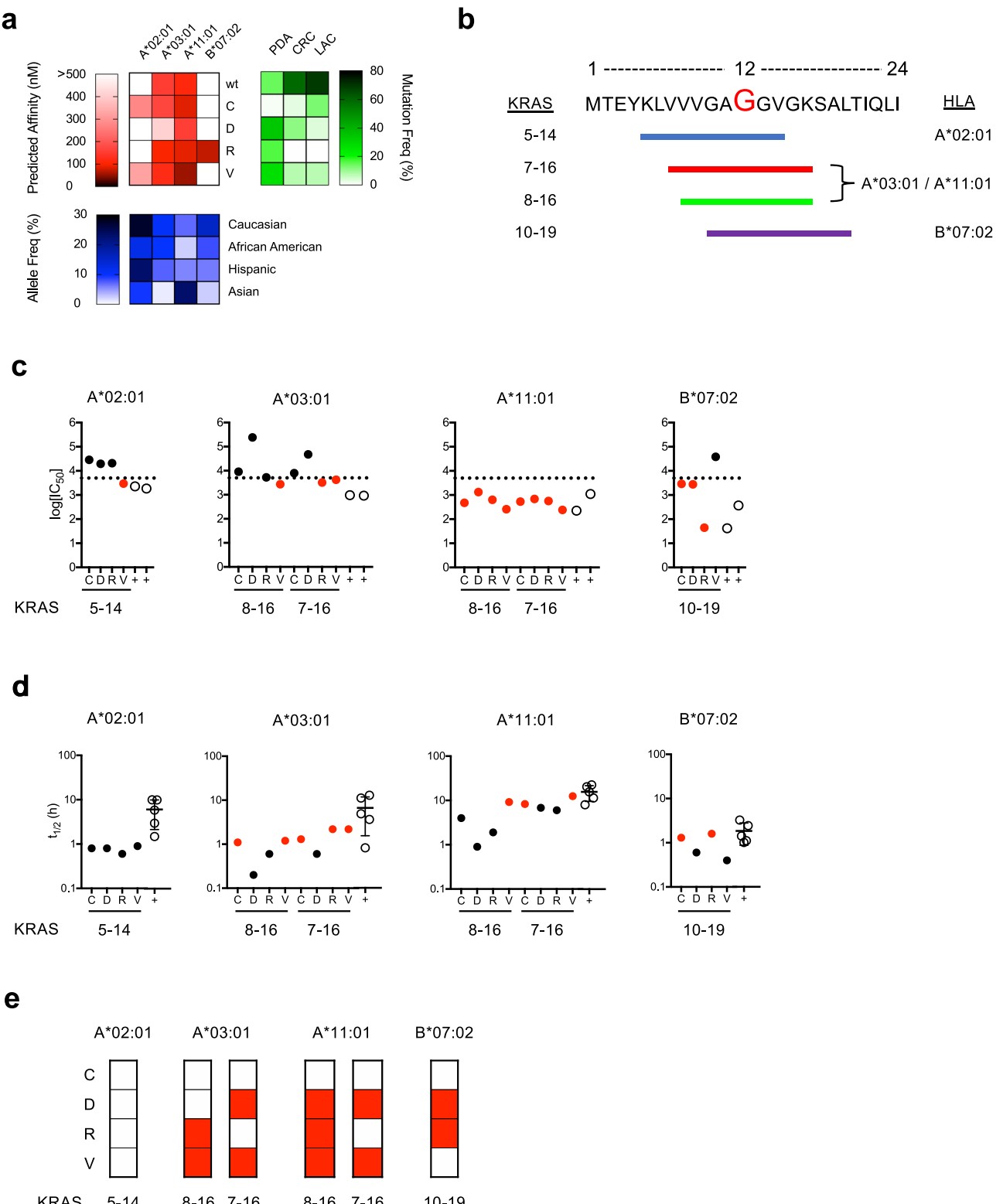

I binding affinity ($\log_{10}[IC_{50}] = 3.469$ nM) (Fig. 1c). However, all p-HLA-A*02:01 class I complexes exhibited short half-lives ($t_{1/2} < 1$ h) compared to control peptides ($t_{1/2}$ 1.3–10.1 h, Fig. 1d). For HLA-A*03:01 epitopes, peptides 7-16R, 8-16V, and 7-16V exhibited strong binding affinity ($\log_{10}[IC_{50}] \leq 3.589$ nM) (Fig. 1c) and formed p-HLA complexes with equivalent stability to positive controls ($t_{1/2} > 1$ h). All 9-mer (8-16C-V) and 10-mer (7-16C-V) peptides bound with strong affinity to HLA-A*11:01 ($\log_{10}[IC_{50}] \leq 3.121$ nM) (Fig. 1c), and

10-mer peptides formed more stable p-HLA-A*11:01 complexes compared to 9-mer peptides ($t_{1/2}$ 6–12 h vs 0.9–9.2 h) (Fig. 1d). For HLA-B*07:02 epitopes, peptides 10-19C, 10-19D, and 10-19R bound with high affinity ($\log_{10}[IC_{50}] \leq 3.460$ nM) (Fig. 1c), but only 10-19C and 10-19R exhibited p-HLA complex stability equivalent to positive control peptides ($t_{1/2} > 1$ h) (Fig. 1d). A comparison of predicted and experimentally determined p-HLA affinities showed a strong correlation for 8-16C-V and 7-16C-V peptides restricted by

**Fig. 1 Identification, characterization, and validation of candidate mKRAS epitopes. a** Heat maps displaying candidate WT and mKRAS epitopes with predicted HLA class I binding affinity (red) to the indicated alleles as determined by antigen.garnish, USA population HLA class I allele frequencies (blue) are as reported in the Allele Frequency Net Database, and mKRAS G12 variant frequencies (green) in pancreatic adenocarcinoma (PDA; $n = 836$), colorectal cancer (CRC; $n = 866$), and lung adenocarcinoma (LAC; $n = 516$) patients are as reported in the ICGC Data Portal. **b** Graphic depiction of the KRAS 24-mer containing candidate epitopes restricted by alleles shown in (**a**). The G12 codon is indicated in red. Epitopes are identified by colored lines, and amino acid positions as well as HLA class I restricting alleles are indicated. **c** Peptide affinity measurements of mKRAS epitopes to HLA-A*02:01, HLA-A*03:01, HLA-A*11:01, and HLA-B*07:02. p-HLA affinities were assessed by fluorescence polarization competitive peptide-binding assays[30]. High peptide affinity is classified as a $\log_{10}[IC_{50}] < 3.7$ nM and is indicated by the dashed line. **d** p-HLA complex stability measurements of mKRAS epitopes to HLA-A*02:01, HLA-A*03:01, HLA-A*11:01, and HLA-B*07:02. p-HLA stability measurements were determined by scintillation proximity assay of β2-microglobulin dissociation[29]. For each HLA class I allele tested, β2-microglobulin association with MHC heavy chain in the absence of peptide served as negative controls. Displayed p-HLA affinity and stability values represent the mean of two independent tests. For affinity and stability assays, validated T cell epitopes for each allele were used as positive controls (open circles) and are listed in Supplementary Table 3. Error bars in panel (**d**) specify mean values ± SD of positive controls. mKRAS peptides with high measured binding affinity or stability are indicated as red circles. **e** Summary chart of MS/MS detected HLA class I restricted mKRAS epitopes processed and presented by TMC-engineered monoallelic cell lines (red). Validated viral epitopes (NLV, ILR, IVT, and TPR) encoded by the TMC construct were used as positive controls for epitope processing and presentation and are listed in Supplementary Table 3. Source data are provided as a Source Data file.

HLA-A*03:01 and HLA-A*11:01 ($R^2 > 0.9$). However, no correlation was observed for either 5-14C-V peptides restricted by HLA-A*02:01 ($R^2 = 0.1033$) or 10-19C-V peptides restricted by HLA-B*07:02 ($R^2 = 0.6945$) (Supplementary Fig. 3 and Supplementary Table 1). A similar analysis of predicted and experimentally determined p-HLA stabilities (Supplementary Table 2) showed no correlation among p-HLA complexes studied. Experimental stabilities for all mKRAS/HLA-A*11:01 complexes yielded higher binding affinities than were predicted. In summary, we experimentally identified 8 mKRAS G12 epitopes restricted to HLA-A*03:01 ($n = 3$), HLA-A*11:01 ($n = 3$), or HLA-B*07:02 ($n = 2$) that display high affinity and formed high-stability p-HLA complexes equivalent to positive control peptides.

**Validation of mKRAS antigen processing and presentation.** To validate epitope processing and presentation, we performed targeted mass spectrometry (MS) on peptides isolated from single HLA class I allele/tandem minigene construct (TMC)-expressing cell lines. HLA class I negative cell lines were engineered to express a GFP-tagged HLA class I/β2-microglobulin single-chain dimer (HLA-SCD) construct and a mCherry-tagged TMC (Supplementary Fig. 4a). Use of monoallelic cells lines eliminates the ambiguity that arises from the coexpression of multiple HLA class I alleles[31]. The TMC consisted of nine minigenes encoding WT and mKRAS long 25-mer peptides (G12C, G12D, G12R, G12V) as well as validated viral epitopes restricted by HLA-A*02:01 (NLV), HLA-A*03:01 (ILR), HLA-A*11:01 (IVT), and HLA-B*07:02 (TPR) as positive controls (Supplementary Fig. 4b and Supplementary Table 3). TMC-expressing monoallelic cell lines were phenotyped for reporter and HLA class I expression (Supplementary Fig. 4c–e) then subjected to HLA class I immuno-precipitation and peptide elution. Peptide sequence identities were determined by quadrupole—orbitrap tandem MS using data-dependent acquisition (DDA) and targeted parallel reaction monitoring (PRM)[32]. Synthetic mKRAS peptides served as "beacons" to zero-in on TMC-encoded peptides that were naturally loaded onto HLA class I molecules. MS/MS fragmentation pattern comparison of eluted and synthetic peptides ensured accurate assignment of AA sequence identity of mKRAS epitopes (Fig. 1e and Supplementary Figs 5–7). In this manner, peptides 7-16D, 8-16V, 7-16V, and 8-16R were eluted in the context of HLA-A*03:01 (Supplementary Fig. 5), and peptides 8-16D, 7-16D, 8-16V, 7-16V, and 8-16R were eluted in the context HLA-A*11:01 (Supplementary Fig. 6). Among these peptides, only 8-16V and 7-16V form p-HLA complexes with HLA-A*03:01 and HLA-A*11:01 with comparable stability to positive controls. Lastly, peptides 10-19D and 10-19R were also identified in the context of HLA-B*07:02 (Supplementary Fig. 7). To further

support the identity assignments between eluted and synthetic peptides, we determined precursor ion similarity via isotopic dot product (idotp, value >0.9) and dot product (dotp, value >0.5) measurements[33]. No mKRAS G12 peptides presented by HLA-A*02:01 were observed despite the detection of the validated NLV viral epitope (Supplementary Fig. 8). In addition, no cysteine-containing epitopes were detected. Free cysteines are prone to post-translational modifications and are underrepresented in MS-based HLA-peptide datasets[34,35]. Future analysis accounting for post-translation modifications will be performed to address the processing of mKRAS cysteine-containing peptides. Importantly, neither 8-16C nor 7-16C in the context of HLA-A*03:01 was observed by Wang et al.[36] suggesting these peptides are not part of this HLA class I peptidome. In sum, we identified 11 processed and presented mKRAS epitopes restricted to HLA-A*03:01 ($n = 4$), HLA-A*11:01 ($n = 5$), or HLA-B*07:02 ($n = 2$) class I alleles (Fig. 1e).

**mKRAS peptide immunogenicity and mKRAS-specific TCRs isolation.** To evaluate the immunogenicity of selected mKRAS peptides, CD8+ T cells isolated from healthy donors were cocultured with autologous monocyte-derived mature dendritic cells (mDC) pulsed with candidate epitopes. Following two stimulations, cultures were screened for CD8+ T cell responses by IFN-γ ELISPOT assay (Fig. 2a). No responses were observed to peptides 5-14C-V among HLA-A*02:01+ donors. Responses were observed to peptide 7-16V in HLA-A*03:01+ ($n = 1/4$) and HLA-A*11:01+ ($n = 4/4$) donors, and responses to 7-16C ($n = 4/4$) and 7-16D ($n = 1/4$) were also noted in HLA-A*11:01+ donors. Responses to peptide 10-19R ($n = 2/5$) in HLA-B*07:02+ donors were also observed.

Given the high sensitivity of TCRs for p-HLA complexes, we sought to isolate TCRs directed against 7-16V and 10-19R epitopes in order to characterize their expression by tumor cell lines and validate these epitopes as neoantigens. To isolate TCRα/β pairs, cell cultures containing CD8+ T cell responses to 7-16V/HLA-A*03:01, 7-16V/HLA-A*11:01, and 10-19R/HLA-B*07:02 were expanded using peptide-pulsed artificial antigen presenting cells (APC). Antigen specificity was confirmed by p-HLA multimer staining (Fig. 2b–d), and multimer+ cells were flow cytometrically sorted to >99% purity. TCRα/β sequences were determined by next-generation DNA and RNA sequencing[37]. A total of five TCRα/β pairs were identified, three of which were selected for further studies: TCRA3V (7-16V/HLA-A*03:01), TCRA11V (7-16V/HLA-A*11:01), and TCRB7R (10-19R/HLA-B*07:02) (Table 1).

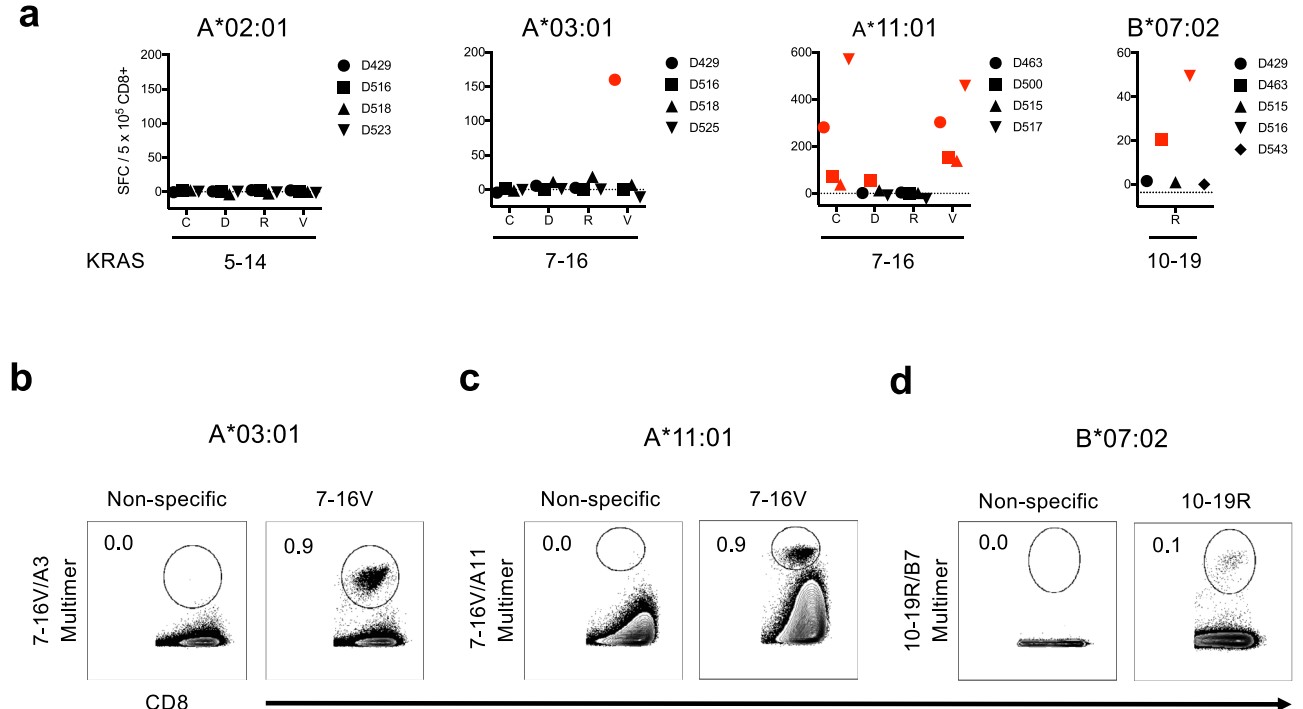

**Fig. 2 Assessment of mKRAS neoantigen immunogenicity and identification of mKRAS-specific TCR sequences. a** IFN-γ ELISPOT assay results following in vitro expansion of healthy donor purified CD8$^+$ T cells stimulated with mKRAS peptide-pulsed autologous mDC. IFN-γ ELISPOT Spot Forming Cell (SFC) values are substrated from media controls for each subject. The data presented are the pooled results from single experiments of individual donors represented by symbols indicated in the figure legend. Red symbols denote donors with positive responses. **b** p-HLA multimer analysis demonstrating positive 7-16V-specific HLA-A*03:01-restricted CD8$^+$ T cell response isolated from donor D429. **c** p-HLA multimer analysis demonstrating positive 7-16V-specific HLA-A*11:01-restricted CD8$^+$ T cell response isolated from donor D500. **d** p-HLA multimer analysis demonstrating positive 10-19R-specific HLA-B*07:02-restricted CD8$^+$ T cell response isolated from donor D516. Cells were stained with p-HLA multimer and anti-CD8 Ab. p-HLA multimer staining of respective donor CD8$^+$ T cells stimulated in the absence of peptide (non-specific) were used as staining controls. A total of 1 × 10$^6$ live events were collected gating on FSC/SSC/CD8$^+$ cells. Data presented in panels (**b**–**d**) are representative of two independent tests. Source data are provided as a Source Data file.

**Table 1 Summary table of mKRAS-TCRs.**

| ID | Epitope | Restriction | Vα | Vβ | CDR3α | CDR3β |
|---|---|---|---|---|---|---|
| TCRA3V | 7-16V | A*03:01 | TRAV19, TRAJ40 | TRBV9, TRBD1, TRBJ2-5 | CALSEAGTYKYIF | CASSVAGGGQETQY |
| TCRA11V | 7-16V | A*11:01 | TRAV12-1, TRAJ8 | TRBV28, TRBD2, TRBJ2-7 | CAVNPPDTGFQKLVF | CASSLSFRQGLREQYF |
| TCRB7R | 10-19R | B*07:02 | TRAV4, TRAJ41 | TRBV7-2, TRBJ1-2 | CLVGDFNSNSGYALNF | CASKVYGYTF |

TCRs were identified following TCRVα and TCRVβ sequencing of flow cytometrically sorted p-HLA+/CD8+ T cells derived from cultures shown in Fig. 2b–d with CDR3 amino acid sequences specified.

We initially characterized the expression and function of these mKRAS-TCRs using a cell reporter system developed in our laboratory designated J$^{ASP90}$ reporter cells. The J$^{ASP90}$ reporter cells are derived from CD8$^+$/TCRαβ null Jurkat E6.1 cells engineered to express NFAT-inducible eGFP as a readout for TCR signaling. mKRAS-TCRα/β chain pairs were cloned into lentiviral vectors and expressed in J$^{ASP90}$ reporter cells. Custom p-HLA multimer staining was performed to validate TCRA3V, TCRA11V, and TCRB7R expression as well as confirm antigen specificity and HLA restriction (Fig. 3a).

To further assess antigen specificity and measure TCR avidity, TCR-engineered J$^{ASP90}$ reporter cells were cocultured with HLA class I matched monoallelic K562 cells pulsed with WT or mKRAS synthetic peptides. TCRA3V, TCRA11V, and TCRB7R-expressing J$^{ASP90}$ reporter cells demonstrated specific reactivity to mKRAS cognate epitopes without reactivity to WT KRAS peptide-pulsed or unpulsed monoallelic K562 cells (Fig. 3b).

Using titrated peptide concentrations, the functional avidities of each TCR were determined based on the mean concentration required to achieve 50% NFAT activation (Fig. 3b). The avidities of TCRA3V, TCRA11V, and TCRB7R were determined to be 1.64, 0.15, and 3.38 nM, respectively. Furthermore, TCRA3V and TCRB7R cells exhibited no cross-reactivity to other mKRAS G12 epitopes, while TCRA11V cells demonstrated cross-reactivity to peptide 7-16C, albeit with decreased avidity as compared to the cognate peptide. These TCRs constituted high-affinity probes for the characterization of naturally processed and presented mKRAS epitopes by human tumor cells.

**mKRAS-specific TCRs confer lytic activity against human tumor cells.** To probe for p-HLA complexes on tumor cell lines, the anti-tumor activity of mKRAS TCRs was evaluated using engineered TCRβ$^{null}$ primary CD8$^+$ T cells. To this end, healthy donor primary CD8$^+$ T cells were gene edited using CRISPR/

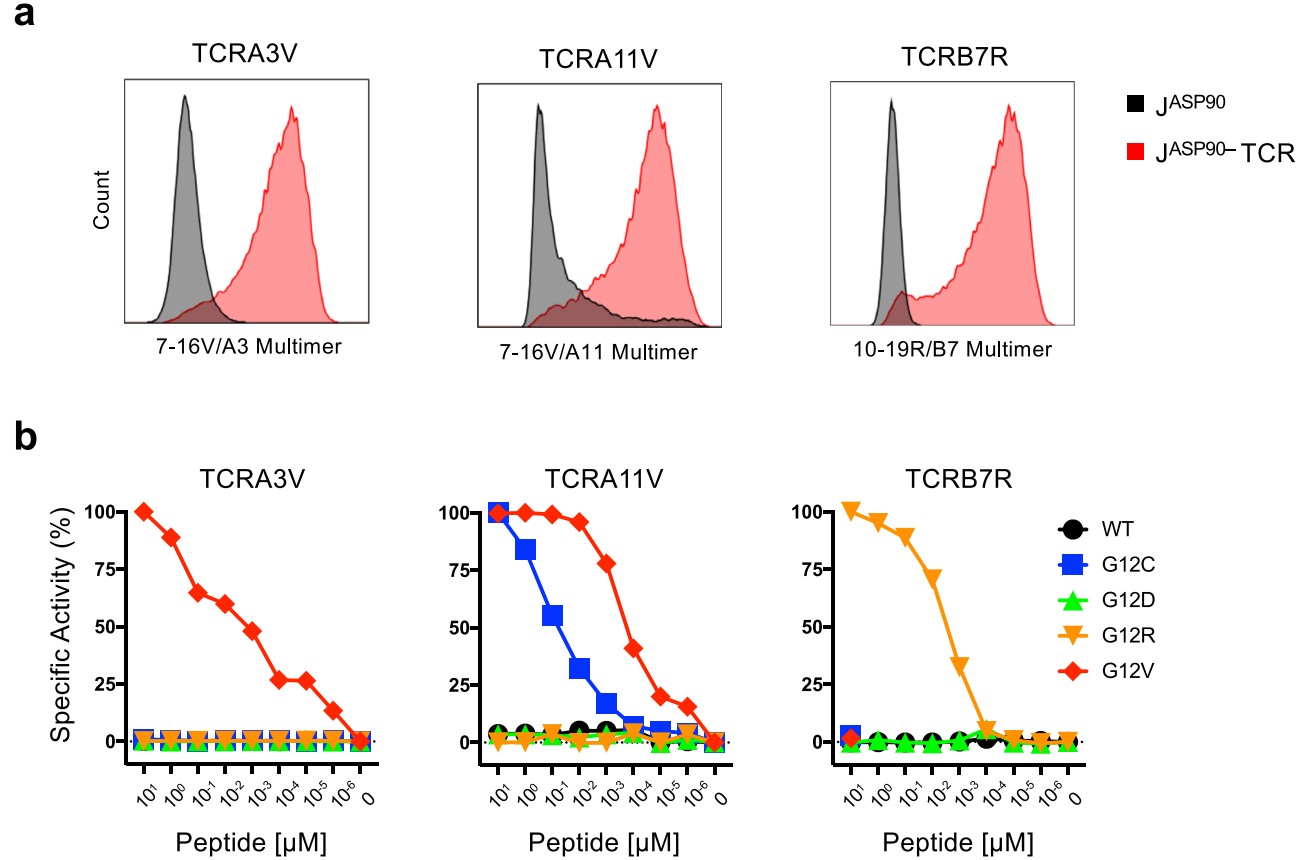

**Fig. 3 Validation and functional characterization of mKRAS-specific TCRs. a** FACS histogram plots demonstrating p-HLA multimer staining following lentiviral engineering of $J^{ASP90}$ reporter cells with TCRA3V, TCRA11V, and TCRB7R constructs (red) vs TCRαβ$^{null}$ controls (black). **b** $J^{ASP90}$ reporter assay to evaluate antigen recognition. $J^{ASP90}$ reporter cells expressing TCRA3V, TCRA11V, or TCRB7R were cocultured with HLA-matched monoallelic K562 cell lines pulsed with titrated concentrations of WT KRAS or mKRAS peptides. TCR signaling is indicated by eGFP expression upon NFAT activation. TCR avidity was determined as the mean peptide concentration required to achieve 50% NFAT activation of TCR-engineered $J^{ASP90}$ reporter cells. Specific Activity (%) = (%GFP$_{Test}$ – %GFP$_{Min}$) / (%GFP$_{Max}$ – %GFP$_{Min}$) x 100. Representative experiments of three independent evaluations are shown. Source data are provided as a Source Data file.

Cas9 and single guide RNAs (sgRNAs) to knock out endogenous TCRα and β expression (Supplementary Fig. 9a–c) to eliminate TCRαβ mispairing and enhance receptor cell surface expression[38,39]. Engineered TCR expression was assessed by cell surface CD3 and TCRαβ staining (Supplementary Fig. 9e). Secondary expansion of mKRAS-TCR T cells using peptide-pulsed artificial APCs led to enrichment of antigen-specific CD8$^+$ T cells (Supplementary Fig. 9f), yielding populations with high (60–90%) expression of TCRA3V, TCRA11V, or TCRB7R as assessed by p-HLA multimer staining (Fig. 4a). In 4 h $^{51}$Cr-release assays, TCRA3V, TCRA11V, and TCRB7R-expressing CD8$^+$ T cells kill HLA class I matched K562 cells pulsed with exogenous cognate mKRAS peptide but not WT KRAS peptide (Fig. 4b). In addition, recognition of processed and presented mKRAS antigen by TCRA3V, TCRA11V, and TCRB7R-expressing CD8$^+$ T cells was demonstrated using TMC-expressing K562 cells as targets (Fig. 4b). Lack of recognition of processed and presented WT KRAS was demonstrated using a WT KRAS-expressing cell line (Fig. 4b, c). Altogether, peptide-pulsing experiments validate antigen specificity while TMC experiments, provide support for proteomic results (Fig. 1e), and validate presentation of cell surface mKRAS p-HLA complexes.

In order to evaluate the recognition of p-HLA complexes on the surface of cancer cells by TCRs, we characterized the expression of HLA class I alleles in a panel of mKRAS G12

cancer cell lines of various tissue origins. Cell lines with reported KRAS WT (TPM 23), G12V (TPM 56-308), and G12R (TPM 16-130) expression, as well as cell lines expressing target HLA class I alleles but alternative KRAS mutations were selected for further study (Supplementary Table 4). Aside from G12V/HLA-A*03:01, no cell lines expressing endogenous G12V/HLA-A*11:01 or G12R/HLA-B*07:02 were identified. When applicable, KRAS G12V$^+$ cell lines were modified to express either HLA-A*03:01 or HLA-A*11:01, and KRAS G12R$^+$ lines were modified to express HLA-B*07:02. As shown in Fig. 4c, d, TCRA3V and TCRA11V-expressing CD8$^+$ T cells exhibited specific cytotoxicity, irrespective of tissue of origin, against all HLA-A*03:01 or HLA-A*11:01-matched KRAS G12V$^+$ cancer cell lines, respectively. Of note, neither TCR conferred cytotoxic activity against HLA matched BxPC-3 tumor cells, a KRAS WT pancreatic adenocarcinoma cell line, validating the specificity of each TCR for the cognate G12V peptide. These observations were replicated and expanded upon using the $J^{ASP90}$ reporter cell system. Upon coculture with mKRAS/HLA-matched cell lines, TCRA3V-, TCRA11V-, and TCRB7R-$J^{ASP90}$ cells demonstrated NFAT-induced GFP expression indicative of TCR sensing of p-HLA complexes presented on the surface of tumor cells (Supplementary Fig. 10a–c). No NFAT-induced GFP expression was observed in $J^{ASP90}$ reporter cells cultured with HLA matched KRAS WT-expressing BxPC-3 cells. The graded pattern of NFAT activation

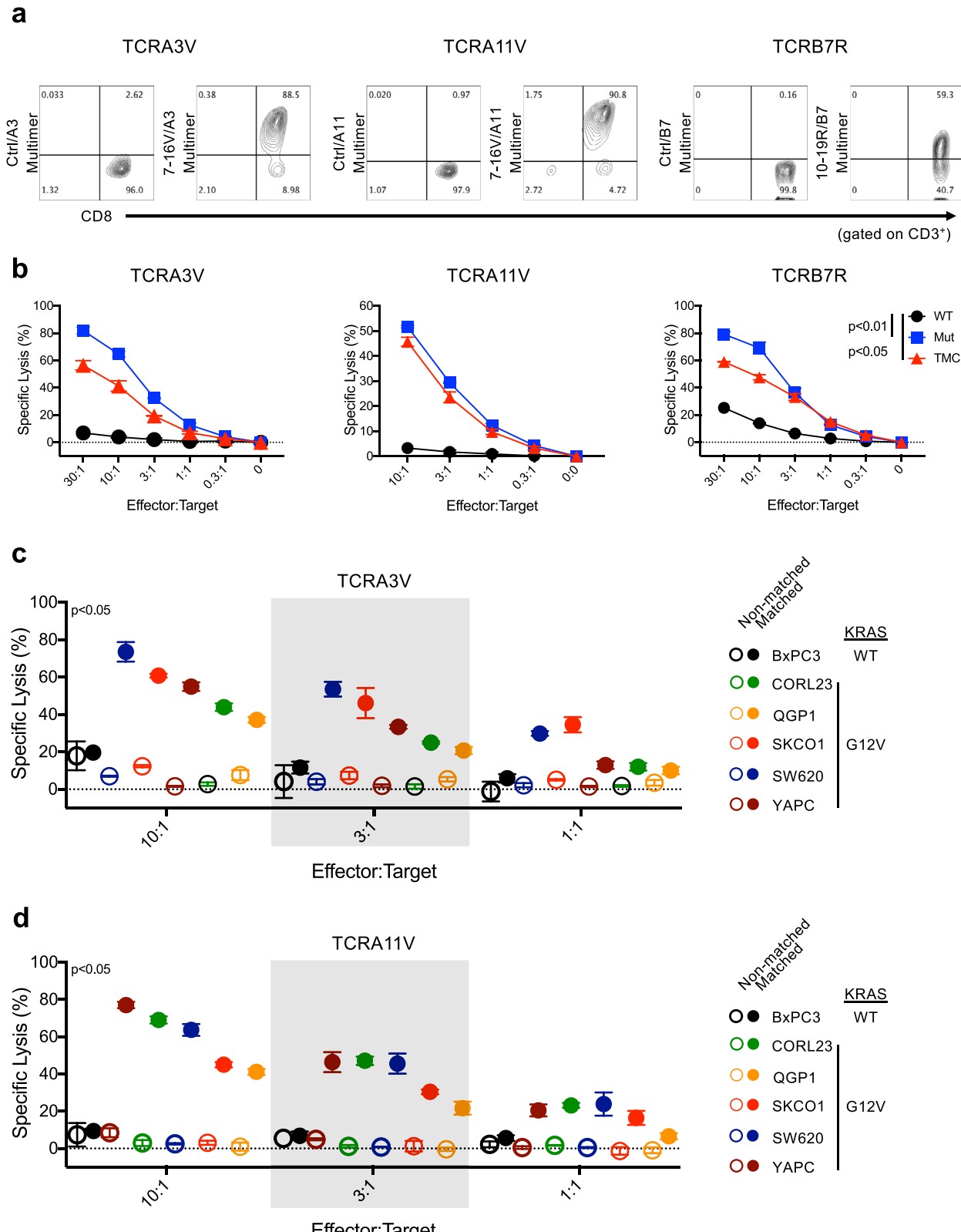

induced in J^ASP90 reporter cells may suggest varied surface expression of p-HLA complexes and/or differing patterns of coreceptor expression by cancer cells. Altogether, these results strongly support the hypothesis that mKRAS epitopes are processed and presented by tumor cells in the context of HLA-A*03:01, HLA-A*11:01 and HLA-B*07:02 at sufficient levels to activate recognition by mKRAS-TCR-engineered T cells.

**Quantitation and recognition of mKRAS peptides presented by human tumor cells**. To gain further insights into the processing and presentation of 8-16V and 7-16V peptides on the surface of cancer cell lines, we performed targeted MS and absolute peptide quantitation to enumerate p-HLA complexes on the surface of CORL23 lung tumor cells engineered to express HLA-A*03:01 (CORL23-A3) or HLA-A*11:01 (CORL23-A11)[40]. CORL23 was

**Fig. 4 mKRAS TCRαβ gene transfer confers HLA-restricted lytic activity against human tumor cells. a** FACS plot demonstrating p-HLA multimer binding to TCRA3V-, TCRA11V-, and TCRB7R-engineered TCRαβ[null] primary CD8[+] T cells gated on viable/CD3[+] population. The following p-HLA multimers were used as staining controls: gp17-25/A3, gp17-25/A11, NY60-72/B7 (Supplementary Table 3). **b** 4 h [51]Cr-release cytotoxicity assay demonstrating TCRA3V, TCRA11V, and TCRB7 cell tumoricidal activity against TMC-expressing (red), mKRAS peptide-pulsed (blue), or WT KRAS peptide-pulsed (black) monoallelic K562 target cells. Percent-specific lysis of triplicates is shown for each data point. Data are presented as mean ± SD. $p < 0.05$ or $p < 0.01$ for all E:T ratios ≥0.3:1 comparing TMC-expressing or mKRAS peptide-pulsed to WT KRAS peptide-pulsed K562 targets, respectively; one-way ANOVA followed by post-hoc pair-wise Student's $t$-test with multiple comparison adjustment. **c, d**. 4 h [51]Cr-release cytotoxicity assay demonstrating TCRA3V and TCRA11V tumoricidal activity against cancer cell lines harboring WT KRAS (BxPC-3) or endogenous KRAS G12V mutation either non-modified (open circles, non-HLA matched) or engineered to express the restricting HLA allele (closed circles, HLA-matched). Each color represents a different cell line as identified in the figure legend. No significant cytotoxicity was observed against HLA matched vs non-matched BxPC-3 cell lines. Data are presented as mean values ± SD ($n = 3$ biologically independent samples). $p < 0.05$ for all E:T ratio ≥1:1 comparing HLA matched vs non-matched cell lines harboring endogenous KRAS G12V mutation; one-way ANOVA followed by post-hoc pair-wise Student's $t$-test with multiple comparison adjustment. Source data are provided as a Source Data file.

chosen for further studies given its ability to recapitulate a model of metastatic lung cancer (see below)[41]. In the context of HLA-A*03:01, we detected 8 complexes of 8-16V and 27 complexes of 7-16V per cell as compared to 43 complexes of 8-16V and 78 complexes of 7-16V per cell in the context of HLA-A*11:01 (Fig. 5a and Supplementary Fig. 11). This result confirms processing and presentation data obtained using TMC-engineered HLA class I monoallelic cell lines (Supplementary Figs 5 and 6) and suggests that KRAS G12V[+] tumors present both 9-mer (8-16V) and 10-mer (7-16V) epitopes in the context of HLA-A*03:01 and HLA-A*11:01. Although these cells were genetically engineered to express the restricting HLA class I allele, HLA-A*03:01 and HLA-A*11:01 cell surface levels were noted to be within the range of expression comparable to other non-engineered HLA-A*03:01[+] and HLA-A*11:01[+] mKRAS tumor cells lines (Supplementary Fig. 12a, b). Importantly, enumerated p-HLA complexes are within the ranges reported for other mKRAS/HLA complexes expressed by non-engineered cancer cell lines[36].

The cytotoxic activity of TCRA3V- and TCRA11V-engineered CD8[+] T cells against CORL23 tumor cell lines was then characterized via live cell imaging and cellular impedance. These complimentary measurements of cell death allow for visualization of the loss of GFP-labeled tumor cells and the accumulation of Annexin V-CF594 dye (Supplementary Fig. 13 and Supplementary Movie 1), as well as cell shrinkage and fragmentation measurements over a 6-day period. TCRA3V cells promoted rapid apoptotic cell death of CORL23-A3 tumor cells at a 10:1 E:T ratio compared to control T cells, with slower killing kinetics at lower E:T ratios of 3:1 and 1:1 (Fig. 5b). TCRA11V cells promoted rapid cell death of CORL23-A11 cells at all E:T ratios (Fig. 5c). The kinetics to reach 50% cytolysis (KT50) was significantly faster for TCRA11V cells compared to TCRA3V cells at all E:T ratios (Fig. 5d), a finding consistent with the higher avidity of TCRA11V (Fig. 3b) and the higher number of 7-16V complexes detected in the context of HLA-A*11:01 expressed on the surface of CORL23-A11 cells.

We next evaluated the sensitivity of mKRAS TCRs to recognize p-HLA complexes in a non-engineered cell line. We identified the LAC cell line NCI-H441 as being both KRAS G12V[+] and HLA-A*03:01[+] (Supplementary Table 4), highlighting it as a potential target of TCRA3V. We confirmed this cell line retained HLA class I expression and expressed low levels of HLA-A*03:01 (Supplementary Fig. 12a). In both 4 h [51]Cr-release and cellular impedance assays, TCRA3V cells exhibited specific lysis against parental (non-engineered) NCI-H441 cells, while TCRA11V cells, which recognizes the identical 7-16V epitope but restricted by HLA-A*11:01, exhibited no cytotoxic activity (Fig. 5e, f). Together, we conclude that TCRA3V and TCRA11V can "sense"

low numbers of p-HLA complexes per cell and represent valuable reagents to immunologically target mKRAS.

**In vivo anti-tumor activity of mKRAS-specific TCR+ CD8+ T cells.** CORL23-A3 and CORL23-A11 tumor cells engineered to express click beetle red (CBR) luciferase were infused into NOD/scid/γcnull (NSG) mice via tail vein injection, and the establishment of pulmonary tumors was confirmed by bioluminescence imaging. Four days post tumor inoculation, mice were treated with TCRA3V- or TCRA11V-engineered CD8[+] T cells. As controls, tumor-bearing mice received either no treatment (Mock) or TCRαβ[null] (TCR KO) CD8[+] T cells. Mice treated with TCRA3V- or TCRA11V- engineered CD8[+] T experienced complete eradication of CORL23-A3 (Fig. 6a, c) or CORL23-A11 (Fig. 6b, d) tumors, respectively, relative to control groups ($p < 0.01$). The treatment effect was long-lasting with no evidence of tumor outgrowth by day 39 in mice treated with mKRAS-TCR therapy, while control mice experienced persistent tumor growth. Consequently, mice treated with mKRAS-TCR therapy experienced prolonged survival relative to control groups ($p < 0.0002$, Fig. 6e, f). Altogether, these results demonstrate that the transfer of T cells engineered with high avidity mKRAS TCRs promotes tumor rejection and prolongs survival.

## Discussion
Epitope discovery is an essential step in designing immunotherapies such as cancer vaccines, bi-specific engagers, and TCR therapies. The ubiquity of mKRAS along with its role as a recurrent clonal driver makes this oncoprotein an ideal target[42–44]. To identify mKRAS epitopes, we undertook a comprehensive multi-omics approach (e.g., bioinformatic, biochemical, proteomic and immunological) to address the intricate processes involved in the generation of immunogenic peptides. By subjecting each candidate peptide to a series of assays that interrogate critical properties of antigen processing and presentation, we have identified a curated, albeit partial set of mKRAS peptides that represent candidate epitopes. Using this approach, we identified three epitopes (7-16V/HLA-A*03:01, 7-16V/HLA-A*11:01, and 10-19R/HLA-B*07:02) that fulfilled criteria of strong measured p-HLA binding affinity, high complex stability, and proteomically validated antigen processing and presentation. Furthermore, we determined that these epitopes were immunogenic allowing for the identification of mKRAS TCR sequences that were used as sensitive probes to validate epitope presentation by human cancer cell lines. Direct quantification of mKRAS p-HLA complexes on CORL23 cells highlights the sensitivity of mKRAS TCRs to recognize low abundance p-HLA complexes and eliminate mKRAS tumors cells in a xenograft mouse model of metastatic cancer.

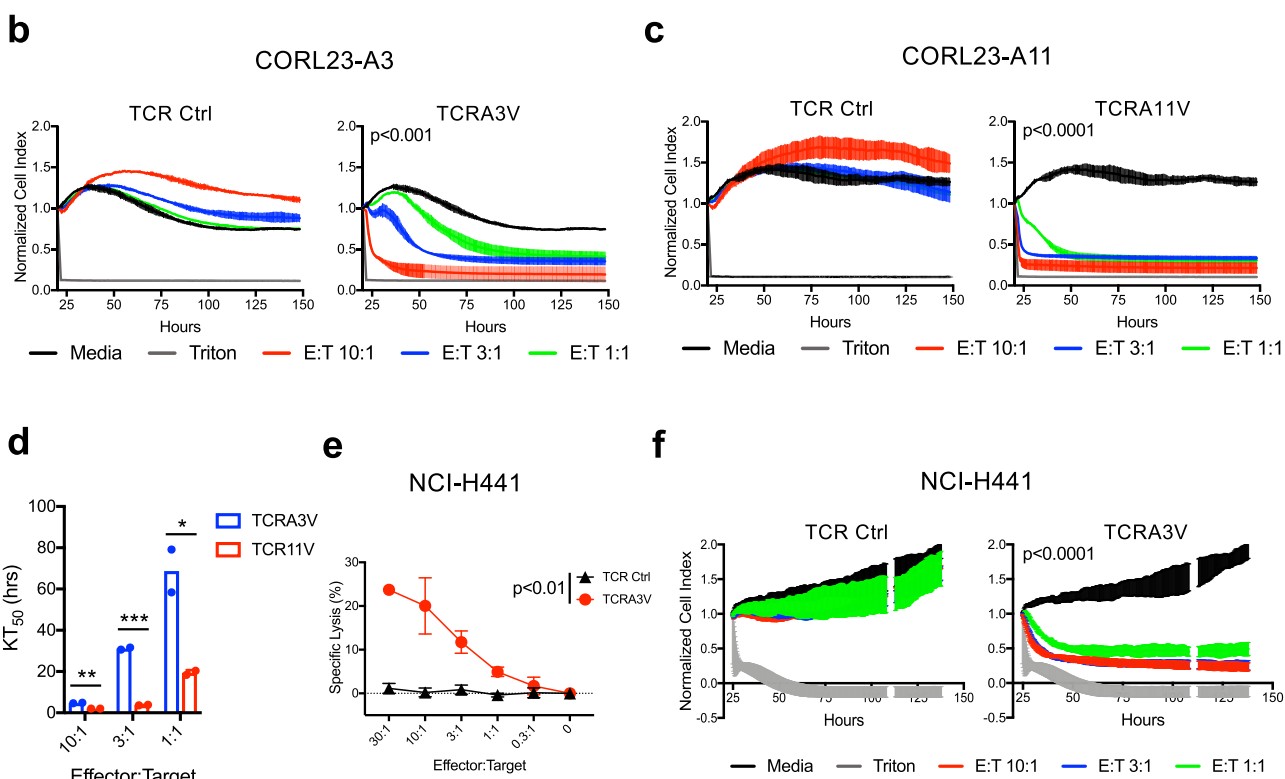

**a**

| COR-L23 | Epitope | Abundance (fM) | Copy Number / Cell |
|---|---|---|---|
| A*03:01 | 8-16V | 4.143 +/- 0.066 | 8 |
| | 7-16V | 14.592 +/- 0.770 | 27 |
| A*11:01 | 8-16V | 23.425 +/- 1.168 | 43 |
| | 7-16V | 42.068 +/- 3.113 | 78 |

**Fig. 5 Quantitation and recognition of KRAS G12V peptide ligands presented by human tumor cell lines. a** Absolute quantification of cell surface expression of 8-16V/HLA and 7-16V/HLA complexes by CORL23-A3 and CORL23-A11 cells. Data are representative of two independent experiments. **b** Cellular impedance to determine the kinetics of tumor cell death upon TCRA3V lytic recognition of CORL23-A3 vs TCR Ctrl (TCRA11V). Normalized cell index values over time for E:T ratios of 10:1, 3:1, and 1:1 are shown compared to tumor cells cultured in media alone. Data for cells exposed to triton-X to represent maximal impedance loss are displayed. Data are presented as mean values ± SD. $p < 0.001$ at 150 h for all E:T ratios for the TCRA3V group compared to media; one-way ANOVA followed by post-hoc pair-wise Student's $t$-test with multiple comparison adjustment. Data are representative of two independent experiments. **c** Cellular impedance data following TCRA11V lytic recognition of CORL23-A11 vs TCR Ctrl (TCRA3V). Data are presented as mean values ± SD. $p < 0.0001$ at 150 h for all E:T ratios for the TCRA11V group compared to media; one-way ANOVA followed by post-hoc pair-wise Student's $t$-test with multiple comparison adjustment. Data are representative of two independent experiments. **d** Comparative kinetics of the time to reach 50% cell death (KT50) of CORL23-A3 and CORL23-A11 cells upon TCRA3V and TCRA11V recognition, respectively, at E:T ratios of 10:1, 3:1, and 1:1. Data are presented as mean values with independent replicates displayed. *$p < 0.05$, **$p < 0.01$, ***$p < 0.001$; multiple $t$-test analysis. Data are representative of two independent experiments. **e** 4 h $^{51}$Cr-release cytotoxicity assay demonstrating TCRA3V tumoricidal activity against NCI-H441 cells (G12V$^+$/HLA-A*03:01$^+$) compared to TCR Ctrl (TCRA11V) cells. Data are presented as mean values ± SD. $p < 0.01$ for all E:T ratio ≥ 1:1; one-tailed Student's $t$-test. Data are representative of two independent experiments. **f** Cellular impedance data following TCRA3V lytic recognition of NCI-H441 vs TCR Ctrl (TCRA11V). Data are presented as mean values ± SD. $p < 0.0001$ at 130 h for all E:T ratios for the TCRA3V group compared to media; one-way ANOVA followed by post-hoc pair-wise Student's $t$-test with multiple comparison adjustment. Data are representative of two independent experiments. Source data are provided as a Source Data file.

Most mKRAS epitopes described herein have not been previously reported as constituents of the HLA peptidome. These epitopes include the following: HLA-A*03:01 – 8-16V, 8-16R, and 7-16V, HLA-A*11:01 – 8-16D, 8-16R, 8-16V, 7-16D, and 7-16V, HLA-B*07:02 – 10-19D and 10-19R. These epitopes exhibit high affinity for the restricting HLA class I allele and relatively high complex stability as each forms p-HLA complexes with $t_{1/2} > 0.5$ h. In agreement with our data, the processing and presentation of the 7-16D peptide in the context of HLA-A*03:01 has also recently been reported by Wang et al.[36] despite exhibiting low measured affinity and complex stability. Our study and those of others[36] using cell lines expressing mKRAS have failed to identify 5-14 mKRAS peptides as constituents of the HLA-A*02:01 peptidome. Furthermore, biochemical assessment of 5-14C-V peptide-binding and stability properties for HLA-A*02:01 suggest that these ligands fail to form stable p-HLA

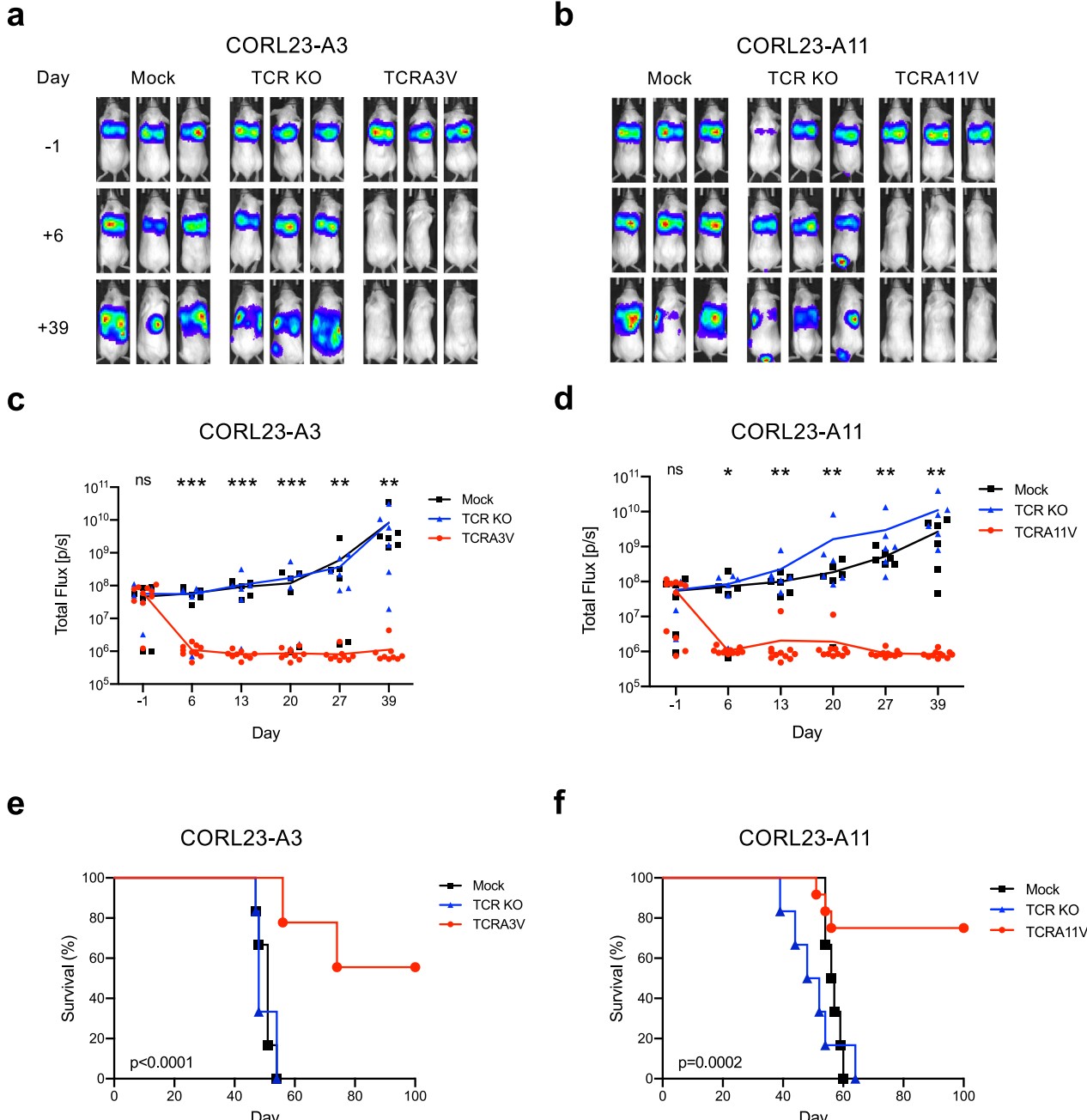

**Fig. 6 Adoptive transfer of mKRAS-TCR T cells leads to in vivo eradication of KRAS G12V+ tumor cells in a xenograft model of metastatic lung cancer.** CORL23-A3 or CORL23-A11 tumors expressing CBR luciferase were engrafted into NSG mice via intravenous tail vein injection. Mice were left untreated (Mock) or treated with TCR αβ null (TCR KO) or mKRAS-TCR engineered CD8+ T cells 4 days after tumor engraftment. Tumor burden was assessed by bioluminescence imaging before and after treatment, and overall survival was monitored over time. Representative bioluminescence imaging prior to and following treatment of NSG mice bearing **a** CORL23-A3 pulmonary tumors treated with TCRA3V T cells and **b** CORL23-A11 pulmonary tumors treated with TCRA11V cells compared to control groups. Total Flux quantification over time of **c** CORL23-A3 and **d** CORL23-A11 tumor-bearing mice as shown in (**a**) and (**b**). Colored lines represent mean Total Flux values over time with individual data points corresponding to treatment groups presented as indicated in the figure legend. *$p < 0.05$, **$p < 0.01$, ***$p < 0.001$; one-way ANOVA followed by Tukey's HSD post-test comparing Mock and TCRA3V or TCRA11V treated mice. No statistical difference was observed between Mock and TCR KO treated mice. Kaplan–Meier analysis of overall survival of **e** CORL23-A3 and **f** CORL23-A11 tumor-bearing mice. Colored lines correspond to treatment groups as indicated in the figure legend. $p$ values as indicated; log-rank testing comparing Mock and TCRA3V or TCRA11V treated mice. No statistical difference was observed between Mock and TCR KO treated mice. Number of mice in representative experiment is as follows: CORL23-A3 Cohort—Mock ($n = 6$), TCR KO ($n = 6$), TCRA3V ($n = 10$). CORL23-A11 Cohort—Mock ($n = 6$), TCR KO ($n = 6$), TCRA11V ($n = 12$). Source data are provided as a Source Data file.

complexes, and only the 5-14V peptide demonstrated high HLA class I affinity in competitive binding assays. By contrast, Mishto et al.[45] reported mKRAS linear (5-14V) and spliced (5-6/8-14V, lacking the V at position 7 of the canonical peptide) peptides as products of in vitro proteasome digests and have demonstrated these peptides to have high affinity for HLA-A*02:01. Given the high prevalence of HLA-A*02:01 in the population, further studies are needed to evaluate the discordant results between processing of the 5-14V mKRAS peptide by cell lines and in vitro proteasome digest assays.

To investigate the immunogenicity of mKRAS epitopes, immune responses were "outsourced" from healthy donors as naïve TCR repertoires may recognize tumor neoantigens[12]. In addition to 7-16V and 10-19R epitopes described, we also identified CD8+ T cell responses to 7-16C and 7-16D in HLA-A*11:01+ donors. Of these epitopes, only 7-16D- and 7-16V-specific TCRs isolated from immunized transgenic HLA-A*11:01 mice have been previously reported[21]. We did not observe immune responses directed against 5-14C-V in HLA-A*02:01+ donors despite previous studies reporting HLA-A2-restricted 5-14D and 5-14V CD8+ T cell responses[17,46]. Given that these studies were reported over 15 years ago, it is likely high-resolution HLA typing was not performed, and it is possible responding patients expressed less prevalent HLA-A2 sub-alleles (e.g., HLA-A*02:03 or HLA-A*02:06) that may exhibit higher binding affinity to 5-14V and 5-14D peptides relative to HLA-A*02:01. Clearly, this issue related to mKRAS epitopes restricted by HLA-A*02:01 requires further investigation.

The availability of TCRs directed against 7-16V/HLA-A*03:01 (TCRA3V), 7-16V/HLA-A*11:01 (TCRA11V), and 10-19R/HLA-B*07:02 (TCRB7R) permitted their use as sensitive probes demonstrating the presence of these p-HLA complexes on the cell surface of various cancer cell lines from multiple histologies. Our observation that NCI-H441 can be recognized by TCRA3V-engineered CD8+ human T cells is noteworthy given this KRAS G12V mutant cell line naturally expresses the HLA-A*03:01 restricting allele. Further insights were gained by the quantification of p-HLA complexes on CORL23 cell lines demonstrating expression of both 9-mer and 10-mer peptides in the context of HLA-A*03:01 and HLA-A*11:01. As TCRA3V and TCRA11V recognize the same 7-16V peptide, we hypothesized differences in TCR avidity as well as p-HLA abundance by CORL23 lines may contribute to the distinct kinetics of in vitro killing. Indeed, higher expression of 7-16V/A*11:01 complexes (78 complexes/cell) relative to 7-16V/A*03:01 (27 complexes/cell) correlates with the more rapid elimination of CORL23-A11 relative to CORL23-A3 tumor cells. Importantly, the adoptive T cell transfer of either TCRA3V or TCRA11V engineered CD8+ human cells efficiently eliminated CORL23 tumor cells in vivo and proved curative for a majority of treated mice.

In summary, epitope identification and validation strategies reliant on a contemporary toolbox of bioinformatic, biochemical, genomic, and immunological assays have recently been employed by several groups in the development of personalized melanoma vaccines targeting passenger mutations[11,47,48]. Here, we have employed a similar strategy to characterize epitopes presented by high prevalence HLA class I alleles targeting a recurrent clonal driver—mKRAS. Our findings provide further evidence highlighting this oncogenic protein as a clinical target of immune-based therapies.

## Methods

**Primary cells.** Peripheral blood mononuclear cells and purified CD8+ T cells were provided by the Human Immunology Core after cell isolation from apheresis products of HLA class I- and class II-typed healthy donors enrolled on the

Institutional Review Board-approved research protocol 705906 at the University of Pennsylvania after providing informed consent.

**Cell lines.** The following cell lines were cultured in RPMI media supplemented with 10% fetal bovine serum (FBS), 2 mM L-glutamine (CORNING 25-005-CI), and 1X penicillin/streptomycin (CORNING 30-002-CI): 721.221 (Fred Hutchinson Cancer Research Center International Histocompatibility Working Group IHW00001), K562 (American Type Culture Collection (ATCC CCL-243)), Jurkat E6-1 (ATCC TIB-152), BxPC-3 (ATCC CRL-1687), CORL23 (Sigma-Aldrich 92031919), NCI-H441 (ATCC HTB-174), SW620 (ATCC CCL-227), YAPC (German Collection of Microorganisms and Cell Cultures (DSMZ) ACC 382), KP-2 (Japanese Collection of Research Bioresources Cell Bank (JCRB) JCRB0181), QGP-1 (JCRB JCRB0183), PSN-1 (ATCC CRL-3211), NCI-H2030 (ATCC CRL-5914), NCI-H358 (ATCC CRL-5807), HuCCT1 (CellBank Australia JCRB0425), and RERF-LC-Ad1 (JCRB JCRB1020). The following cell lines were cultured in DMEM media supplemented with 10% FBS, 2 mM L-glutamine, and 1X penicillin/streptomycin: SK-CO-1 (ATCC HTB-39) and HuP-T3 (DSMZ ACC 259). The following cell lines were cultured in EMEM media supplemented with 10% FBS, L-glutamine, and penicillin/streptomycin: CAL-62 (DSMZ ACC 448) and PANC1 (Laboratory of Dr Gregory Beatty at the University of Pennsylvania).

**Antibodies and flow cytometry.** The following primary antibodies along with vendor, catalog number, and working dilution used for flow cytometry in this manuscript are as follows: anti-human HLA-A, B, C (W6/32)—APC (BioLegend, 311410, 1:50), anti-human HLA-A*02:01—PE (BD, 558570, 1:50), anti-human HLA-A*03:01—Biotin (One Lambda, BIH0269, 1:20), anti-human HLA-A*11:01—Biotin (One Lambda, BIH0084, 1:20), anti-human HLA-B*07:02—PE (BioLegend, 372403, 1:50), anti-human CD3—FITC (Invitrogen, MHCD0301, 1:100), anti-human CD8—APC (ThermoFisher Scientific, MHCD0805, 1:100), anti-human CD8 (SK1)—APC (BioLegend, 344722, 1:100), anti-human TCRαβ (IP26)—APC (BioLegend, 306717, 1:20), and anti-human TCRαβ (IP26)—Biotin (BioLegend, 306703, 1:20). The following custom p-HLA multimers along with vendor and working dilution information are as follows: VVVGAVGVGK/HLA-A*03:01—PE (Immudex, Custom, 1:100), VVVGAVGVGK / HLA-A*11:01—PE (Immudex, Custom, 1:100), GARGVGKSAL/HLA-B*07:02—PE (Immudex, Custom, 1:100). Cells were stained at the specified antibody dilutions and washed in FACS buffer. Data were acquired with a BD LSRFortessa flow cytometer using BD FACSDiva software (version 8.0.2) and analyzed using FlowJo software (version 10.0.7). ELISPOT assays utilized anti-human IFN-γ 1-D1K (Mabtech, 3420-3-250, 1:100), and anti-human IFN-γ 7-B6-1—Biotin (Mabtech, 3420-6-1000, 1:1000). For rapid expansion of sorted T cells, anti-human CD3 (OKT3) (LEAF, 317336, 30 ng/ml) was used. For HLA class I immunoprecipitation, anti-human HLA-A, B, C (W6/32)—AC (Santa Cruz Biotech, sc-32235 AC, 10 μg per 1 mg protein lysate) was used.

**Computational prediction of mKRAS G12 neoepitopes.** We utilized the tumor neoantigen prediction and multi-parameter quality analysis tool antigen.garnish[24] to identify HLA class I binding peptide ligands of mKRAS G12. The input file consisted of the amino acids 1–24 of WT KRAS as well as the most prevalent mKRAS somatic G12 variants (G12C, G12D, G12R, and G12V) screened against HLA class I alleles with a prevalence >1% of the USA population. Nonamer and decamer peptide ligands with ensemble predicted binding affinities <500 nM to target HLA class I molecules were designated candidate mKRAS binding ligands. HLA prevalence among Caucasians, African Americans, Hispanics, and Asians represented in the USA population were gathered from the Allele Frequency Net Database[49]. WT and mKRAS G12 somatic variant frequencies were collated from The International Cancer Genome Consortium Data Portal utilizing pancreatic adenocarcinoma (Project: PAAD-US, n = 177; PACA-CA, n = 268; PACA-AU, n = 391), colorectal cancer (Project: COAD-US, n = 402; READ-US, n = 321; COCA-CN, n = 321), and LAC (Project: LUAD-US, n = 516) patient cohorts.

**Peptides and biochemical peptide-MHC binding assays.** Custom peptides were synthesized by New England Peptide (Gardner, MA) to >95% purity. Lyophilized peptides were dissolved in 10% DMSO (Mylan Cryoserv 67457-178-50) in sterile water pH 7.4 and passed through a 0.2 uM Centrex filter (10467013). Peptide binding to HLA class I alleles was evaluated using a fluorescence polarization-based competition assay (Pure Protein, L.L.C.) to determine the half maximal inhibitory concentration (IC₅₀) of test peptides as a measure of immunogenic potential. The affinity scale for this assay is as follows: High = $\log_{10}[IC_{50}] < 3.7$ nM, Intermediate = $\log_{10}[IC_{50}] = 3.7$–4.7 nM, Low = $\log_{10}[IC_{50}] = 4.7$–5.5 nM, and Very Low = $\log_{10}[IC_{50}] = 5.5$–6.0 nM[30]. p-HLA complex stability measurements were determined using a scintillation proximity assay with association/dissociation curves fitted using GraphPad Prism version 7. In a random sample of 384 peptides, the scintillation proximity assay could discriminate among peptides with affinities <500 nM, stable complexes with half-life between 0.1 and 46 h[50].

**HLA class I genotyping and phenotyping of cell lines.** Tumor cell line HLA class I typing data were obtained using the TRON Cell Line Portal[51] when available. HLA class I typing for CORL23 and CAL-62 cell lines was performed by standard

molecular typing (Hospital of the University of Pennsylvania, Philadelphia, PA). HLA haplotypes of cell lines are presented in Supplementary Table 4. Tumor cell lines were transduced with lentiviral particles expressing HLA class I/β2-micro-globulin single-chain dimer constructs (HLA-SCD, eGFP+) to generate mKRAS G12/HLA class I-matched cell lines. HLA class I cell surface expression was assessed by flow cytometry using APC-conjugated anti-human HLA-A, B, C (clone W6/32). HLA-allele-specific antibodies conjugated to PE, APC, or biotin were used to assess target HLA expression.

**Lentiviral vector constructs and production**. All lentiviral constructs were generated using the third generation lentiviral transfer vector pTRPE-eGFP-T2A-mCherry (generously provided by Dr Michael C. Milone, University of Pennsylvania, Philadelphia, PA). The HLA-A*02:01-SCD was PCR amplified from a HLA-A*02:01-SCD plasmid[52] and used to replace the mCherry moiety on pTRPE-eGFP-T2A-mCherry vector backbone. All other HLA class I-SCD constructs were generated by exchanging HLA-A*02:01 with HLA-A*03:01, HLA-A*11:01 or HLA-B*07:02. To generate the TMC, eGFP within the lentiviral transfer vector backbone was replaced with ubiquitin to generate pTRPE-Ubiquitin-T2A-mCherry. A synthetic DNA vector construct was synthesized (TWIST Bioscience, San Francisco, CA) encoding the first 25 amino acids of WT and mKRAS (G12C, G12D, G12R, G12V) expressed in tandem without linker sequences (Supplementary Fig. 4). The TMC also included viral epitopes embedded within natural 20 amino acid viral protein sequences restricted to HLA-A*02:01 (NLVPMVATV), HLA-A*03:01 (ILRGSVAHK), HLA-A*11:01 (IVTDFSVIK), and HLA-B*07:02 (TPRVTGGGAM). The TMC was PCR amplified and inserted between ubiquitin and T2A sequences. Synthetic TCR DNA vector constructs were synthesized (TWIST Bioscience, San Francisco, CA) to include the TCRα chain followed by the TCRβ chain separated by a T2A sequence. TRAC and TRBC regions of synthetic TCRs were codon altered to make them resistant to Cas9 protein riboprobes targeting endogenous TRAC and TRBC1/TRBC2. TCR constructs were PCR amplified and cloned into a lentiviral vector backbone. High titer lentiviral vector production was performed[53,54].

**Monoallelic KRAS–tandem minigene construct (TMC) cell lines**. HLA class I monoallelic cell lines were generated by transduction of K562 or 721.221 cell lines using lentiviral particles encoding HLA-SCD/eGFP[52]. Cells were subsequently transduced with lentiviral particle encoding TMC/mCherry then sorted to >99% purity by flow cytometry based on coexpression of GFP, mCherry, and HLA class I expression as measured by APC-conjugated anti-human HLA-A, B, C (clone W6/32) staining.

**Proteomic validation of mKRAS antigen processing and presentation**
*Sample preparation*. Monoallelic KRAS-TMC K562 and 721.221 cell lines were expanded to 1–2 × 10^8 total cells and HLA class I immunopurification (IP) was performed[31] using MHC class I (W6/32) antibody non-covalently linked to agarose beads (Santa Cruz Biotechnology, Dallas, TX). Peptides were eluted from HLA class I molecules using 0.1% trifluoroacetic acid (TFA). Immunoprecipitation eluent was passed through a 10,000 Da Amicon molecular weight cut off filter (Merck Millipore) via centrifugation at 10,000 × g for 10 min. Filtered eluent was dried and resuspended in 100 µl of 0.1% TFA. Stage tip C18 columns (Harvard Apparatus) were equilibrated with 200 µl of acetonitrile and 200 µl of 0.1% TFA, and samples were loaded onto columns for desalting. Samples were then washed with three rounds of 200 µl of 0.1% TFA then eluted in 70% acetonitrile in 0.1% formic acid (FA) and dried.

*Data-dependent acquisition (DDA)*. DDA analysis of monoallelic KRAS-TMC K562 and 721.221 cell lines was performed by the Quantitative Proteomics Resource Core at the University of Pennsylvania. DDA experiments were carried out using half of each enriched sample by nano LC-MS/MS using an Orbitrap Fusion (Thermo Scientific) coupled to an Easy-nLC system (Thermo Scientific). Samples were resuspended in 10 µl of 0.1% TFA and 5 µl was injected for each analysis. Samples were separated with an in-house packed column with ReproSil-Pur C18 AQ 3 µm resin with dimensions 75 µm × 20 cm (Dr Maisch GmbH, Ammerbuch, Germany) at a flow rate of 400 nl/min. Using 0.1% FA as buffer A and 80% ACN 0.1% FA as buffer B, peptides were eluted with a gradient of 4% buffer B to 35% buffer B in 50 min, then to 65% buffer B in 5 min, and finally a 5 min wash at 95% buffer B. Peptides were ionized in a Nanospray Flex Ion Source (Thermo Scientific) at 2.3 kV. An MS1 scan was acquired using a resolution of 120,000 FWHM, AGC target of 4e5, and maximum inject time of 50 ms. Precursors were isolated with 1.6 m/z isolation window and fragmented with an HCD collision energy of 30 eV. An MS2 scan was acquired using a resolution of 30,000 FWHM, AGC target of 5e4, and maximum ion inject time of 54 ms in 3 s cycle time scan mode. DDA data were processed with PEAKS X+ version 10.5 with 10 ppm parent mass tolerance, 0.02 Da fragment mass error tolerance, and no enzyme specificity. Data were searched against the SwissProt Human proteome appended with the custom KRAS peptide sequences.

*Parallel reaction monitoring (PRM)*. PRM analysis of monoallelic KRAS-TMC K562 and 721.221 cell lines was performed by the Quantitative Proteomics

Resource Core at the University of Pennsylvania. Targeted Analysis by PRM was performed on the remaining half of each sample to increase the sensitivity of mKRAS epitope detection. Samples were analyzed on a Q-Exactive HF-X (Thermo Scientific) coupled to an Ultimate 3000 nano UHPLC system (Thermo Scientific) using a PRM strategy[32]. Samples were resuspended in 10 µl of 0.1% TFA and 5 µl was injected for each analysis. Samples were separated with an in-house packed column with ReproSil-Pur C18 AQ 3 µm resin with dimensions 75 µm × 20 cm (Dr Maisch GmbH, Ammerbuch, Germany) at a flow rate of 400 nl/min. Using 0.1% FA as buffer A and 80% ACN 0.1% FA as buffer B, peptides were eluted with a gradient of 4% buffer B to 30% buffer B in 42 min, then to 65% buffer B in 6 min, followed by a 7 min wash at 95% buffer B, and a 5 min equilibration at 4% buffer B. Peptides were ionized in a Nanospray Flex Ion Source (Thermo Scientific) at 2.3 kV. An MS1 scan was acquired using a resolution of 120,000 FWHM, AGC target of 1e5, and maximum inject time of 50 ms. An inclusion list of 33 unique ions was used to sequentially isolate and fragment each target peptide with an isolation window of 1.6 m/z and an HCD collision energy of 28 eV. An MS2 scan was acquired using a resolution of 15,000 FWHM, AGC target of 1e5, maximum ion inject time of 100 ms, and loop count of 10. Results were analyzed with Skyline[55] (version 20.1) and reference spectra of synthetic peptides (New England Peptide, Gardner, MA) were used to validate fragmentation patterns. Spectra were manually annotated with IPSA[56]. Synthetic peptides were analyzed on a Q-Exactive HF (Thermo Scientific) coupled to an Ultimate 3000 nano UHPLC system (Thermo Scientific) with DDA. Each synthetic peptide was injected at 1 pmol/µl and analyzed with similar liquid chromatography conditions. An MS1 scan was acquired at a resolution of 120,000 FWHM, AGC target of 3e6, maximum inject time of 32 ms. The top 20 intense ions were isolated and fragmented with a dynamic exclusion of 45 s. For each peptide, an MS2 scan was acquired using a resolution of 15,000 FWHM, AGC target of 2e5, maximum ion inject time of 32 ms, and isolation window of 1.4 m/z. Ions were filtered for charges 2–5. Raw files were searched using Proteome Discoverer 2.2 (Thermo Scientific) against a database of targets with non-tryptic digestion, precursor mass tolerance of 10 ppm, and fragment mass tolerance of 0.02 Da. Search results were filtered with the target decoy approach[40]. The validation of the precursor ion identity was based on isotopic dot product (idotp) and dot product (dotp) values generated by Skyline[33]. Both values provide a measurement of similarity between eluted and expected or reference peptide. The idotp value provides a measure to assess the precursor isotope distribution and its correlation between observed (eluted) and expected (synthetic) peptide while the dotp value provides a measure of similarity between observed and library spectrum (Proteome Discoverer Database) peak areas. Values range from a best of 1.0 to a worst of 0.0.

**Proteomic quantitation of mKRAS peptides presented by human tumor cells**
*Sample preparation*. CORL23-A3 and CORL23-A11 cells were expanded to 1–2 × 10^8 total cells. HLA class I IP was performed by Cayman Chemical (Ann Arbor, MI). Cells were resuspended in MHC lysis buffer consisting of DPBS (0.25 % Sodium deoxycholate, 200 µM iodoacetamide, 1% N-Octyl-β-D- thioglucoside, 1 mM EDTA) containing protease inhibitor (1 ml of lysis buffer per 1 × 10^8 cells). Cell pellets were resuspended by pipetting and vortexing six times for 3–4 s each at 5-min intervals. Lysate was centrifuged at 800 × g for 5 min and the supernatant was transferred to a 50-ml tube. Additional lysis buffer was added to the supernatants (to double the volume) prior to centrifugation at 20,000 × g for 60 min. The supernatants were transferred to a new 15 ml conical tube. Then, 200 µl from each tube was saved for analysis (Pre-IP). The cell lysate was then mixed with 100 µl of W6/32 resin and incubated overnight at 4 °C with gentle rotation. The IA resin (W6/32 antibody DMP crosslinked to protein A resin) was washed with lysis buffer before use. The mixture was centrifuged at 800 × g for 5 min and the supernatant was transferred to a new tube (500 µl of the supernatant (Post-IP) was retained for analysis). The IA resin pellets were washed in 2.5 ml of lysis buffer, centrifuged at 800 × g for 5 min at 4 °C, and the supernatant was aspirated. Two additional washes were carried out using 2.5 ml each of wash buffers 2 (20 mM Tris-HCL, 400 mM Nacl, pH 8.0) and 3 (20 mM Tris-HCL, 150 mM Nacl, pH 8.0). The IA resin was transferred to a low protein binding Eppendorf tube for a final wash. 0.75 ml of wash buffer 4 (20 mM Tris-HCL, pH 8.0) was added, mixed, centrifuged at 800 × g for 5 min and the supernatant was aspirated. Peptides were eluted from MHC class I molecules using 1 ml of MHC-I elution buffer (0.1 M Acetic acid, 0.1% TFA), incubated for 5 min at 37 °C and centrifuged at 800 × g for 5 min at 4 °C. The eluate was collected into fresh low protein binding tubes, and was centrifuged at 20,000 × g for 2 min to clear the sample of any remaining resin. The eluate was collected into a new low protein binding tube, flash frozen in liquid nitrogen, and stored at −80 °C. A quality assurance ELISA assay was performed to quantify the enrichment of HLA class I protein in Pre- and Post-IP samples.

*Data-dependent acquisition (DDA)*. DDA analysis of CORL23 cell lines was performed by MS Bioworks (Ann Arbor, MI). DDA experiments were carried out using half of each enriched sample by nano LC-MS/MS using a Waters M-Class system interfaced to a ThermoFisher Fusion Lumos mass spectrometer. Peptides were loaded on a trapping column and eluted over a 75-µm analytical column at 350 nl/min; both columns were packed with Luna C18 resin (Phenomenex). A 2 h reverse phase gradient was employed. The mass spectrometer was operated in a combined data-dependent EThcD/CID mode, with MS and MS/MS performed in

the Orbitrap at 60,000 FWHM resolution and 15,000 FWHM resolution, respectively. The instrument was run with a 3-s cycle for MS and MS/MS. DDA data were processed with PEAKS X+ version 10.5 with 10 ppm parent mass tolerance, 0.02 Da fragment mass error tolerance, and no enzyme specificity. Data were searched against the SwissProt Human proteome appended with the custom KRAS peptide sequences. Peptide 8-16V (VVGAVGVGK) and 7-16V (VVVGAVGVGK) identifications were confirmed with analysis of synthetic peptide standards (New England Peptide).

*Quantification of mKRAS epitopes by parallel reaction monitoring (PRM).* The quantification of mKRAS epitopes expressed by HLA-engineered CORL23 cell lines was performed by targeted MS using PRM (MS Bioworks, Ann Arbor, MI). Peptides were enriched as described above. Synthetic stable labeled peptides VVGAVGVGK^ and VVVGAVGVGK^ were purchased from New England Peptide, where ^ is Lysine ($^{13}C_6$$^{15}N_2$). In all, 200 fmols of each stable labeled peptide was added to the enriched samples for analysis. Each enriched sample was analyzed in duplicate (50% of the sample per injection). PRM was performed with a Waters M-Class HPLC system interfaced to a ThermoFisher Fusion Lumos mass spectrometer. Peptides were loaded on a trapping column and eluted over a 75-μm analytical column at 350 nl/min; both columns were packed with Luna C18 resin (Phenomenex). The mass spectrometer was operated in PRM mode with the Orbitrap operating at 17,500 FWHM resolution. Collision-induced dissociation data were collected for the $(M+2H)^{2+}$ charge state ions of the target peptides VVGAVGVGK, VVGAVGVGK^, VVVGAVGVGK, and VVVGAVGVGK^. Extracted ion chromatograms for each of the target peptides were generated manually using the XCalibur QualBrowser software version 4.1.31.9 (Thermo-Fisher) with a 20 ppm mass tolerance for product ions. For the calculation of 8-16V and 7-16V epitopes expressed by HLA-A*03:01 and HLA-A*11:01 complexes, eluted and internal standard peptide peak area data were used to calculate the number of moles of peptide present in the sample. This number was doubled (half the IP was analyzed in a single injection) and converted to molecules by multiplying by Avogadro's number. The result was divided by the number of input cells (164M and 163M for the CORL23-A3 and CORL23-A11 cell lines, respectively) to give number of peptide molecules/cell.

**Generation of mKRAS-reactive T cell lines.** The generation of monocyte-derived mDC and antigen-specific T cells was performed[57]. Briefly, DC were generated from monocyte cultures in GM-CSF (100 ng/ml, Berlex) and IL-4 (20 ng/ml, Miltenyi Biotec) and matured using CD40L-expressing K562 cells, IFN-γ (100 u/ml, CellGenix), poly I:C (Invivogen, Inc), and R848 (Invivogen, Inc.) for 16 h to generate mDC. CD8+ T cells were isolated from peripheral blood mononuclear cells, using a CD8 negative selection kit (Miltenyi Biotech, Auburn, CA). Purified CD8+ T cells ($5 \times 10^6$ cells/ml) were cultured at a 20:1 ratio with irradiated (2500 Rads) autologous mDC pulsed with peptide (40 μg per $1 \times 10^6$ DC/ml) in 24-well trays in Optimizer CST media (Gibco) supplemented with 5% pooled human sera. Human IL-7 (10 ng/ml), IL-15 (5 ng/ml), and IL-12 (10 ng/ml) were added on day 0. Fresh media supplemented with IL-7 (10 ng/ml) and IL-15 (5 ng/ml) was added on day 7. Fourteen days after primary mDC stimulation, T cell cultures were harvested and re-stimulated with irradiated (2500 Rads) peptide-pulsed mDC. Cell culture media was supplemented with 50 U/ml IL-2 (Chiron, Emeryville, CA) starting day 2 then every 48 h following secondary stimulation. On days 10–14 of secondary mDC stimulation, antigen-specific T cell responses were identified by IFN-γ ELISPOT assay and flow cytometry by p-HLA multimer staining.

**IFN-γ ELISPOT assay.** CD8+ T cell reactivity to peptide antigen was assessed by interferon-γ (IFN-γ) ELISPOT assay[58]. The spot number was determined in an independent blinded fashion (ZellNet Consulting, New York, NY) using the high-resolution automated KS ELISPOT reader (Zeiss, Thornwood, NY) and KS ELISPOT 4.9.16 software with reading parameters established per International Harmonization Guidelines[59]. A positive response was recorded if the number of spots in the peptide-exposed wells was two times or more higher than the number of spots in the unstimulated wells and if there was a minimum of 20 (after subtraction of background spots) peptide-specific spots per $5 \times 10^5$ CD8+ cells.

**p-HLA multimer assay.** mKRAS-specific CD8+ T cell frequencies were determined by staining with PE-conjugated p-HLA dextramers (Immudex), followed by addition of APC-conjugated anti-human CD8 antibody (Invitrogen). Cells were washed and resuspended in FACS buffer. One million events in the CD8+ gate were collected using a hierarchical gating strategy. Data were acquired using BD FACSDiva software (version 8.0.2) and analyzed using FlowJo software (version 10.7.2).

**Isolation and validation of KRAS-specific TCRs.** mKRAS-specific CD8+ T cell cultures underwent antigen-specific expansion using irradiated (10,000 Rads) HLA-SCD/4-1BBL expressing K562 artificial APC at a 1:1 ratio. Cultures were supplemented with IL-2 (500 U/ml) 48 h after initiation then every 48 h thereafter until culture termination. T cells were expanded for 12–14 days, sorted to 97–99% purity by CD8 and p-HLA multimer coexpression.

*TCRα/TCRβ sequencing.* DNA was extracted using the Gentra Puregene cell kit following the manufacturer's directions (Qiagen, Valencia, CA, Cat. No. 158388). Bulk DNA TCR Vβ and Vα libraries were prepared for sequencing on the Illumina MiSeq platform using a cocktail of 23 Vβ families from framework region 2 (FR2) forward primers, and 13 Jβ region reverse primers, modified from the BIOMED2 primer series[60] and a cocktail of 39 Vα from FR3 primers and 20 Jα region reverse primers, respectively. Libraries were generated using the QIAGEN Multiplex PCR and Illumina Nextera XT index kits[61]. RNA was extracted using RNeasy Plus Mini Kit following the manufacturer's directions (Qiagen, Cat. No. 74134). Bulk RNA TCR Vβ and Vα libraries were prepared using SMARTer® Human TCR α/β Profiling Kit following the manufacturer's directions (Takara Bio USA, Mountain View, CA, Cat. No. 635014). Single cells were sorted into a 96-well plate and single cell sequencing libraries were prepared using SMARTer® Human scTCR a/b Profiling Kit following the manufacturer's directions (Takara Bio USA, Inc., Cat. No. 634431). Libraries were sequenced ($2 \times 300$ bp paired end reads, MiSeq Reagent Kit v3-600 cycle, Illumina, San Diego, CA, USA; Cat. No. 102-3003) on an Illumina MiSeq instrument in the Human Immunology Core Facility at the University of Pennsylvania.

*Data analysis.* Raw sequences were quality filtered[62] and clone assemblies were processed with MiXCR (v. 3.0.7)[63] and VDJtools (v1.2.1)[64].

**Jurkat reporter system to assess TCR antigen specificity and avidity.** Using Cas9 protein and riboprobes as guides against *TRAC* (5'-TGTGCTAGACATGA GGTCTA-3') and *TRBC1/TRBC2* (5'-GGAGAATGACGAGTGGACCC-3')[38], a Jurkat E6.1 TCRα/β-negative cell line was generated. To generate J$^{ASP90}$ reporter cells, TCR-negative Jurkat E6.1 cells were then transduced with lentiviral particles encoding human CD8αβ heterodimer (generously provided by Dr Jim L. Riley, University of Pennsylvania, Philadelphia, PA) and the Uni-Vect reporter system that features constitutive mCherry and NFAT-inducible eGFP expression as a means to assess TCR signaling (generously provided by Dr Daniel J. Powell Jr., University of Pennsylvania, Philadelphia, PA). J$^{ASP90}$ reporter cells were flow cytometrically sorted to purity based on TCR⁻ (anti-human TCRα/β antibody clone IP26, BioLegend, San Diego, CA), CD8+ (anti-CD8 antibody, ThermoFisher Scientific, Waltham, MA), mCherry+ expression, and low basal NFAT (eGFP+) signal. The TCRα and TCRβ chains of TCRA3V, TCRA11V, and TCRB7R were expressed via lentivirus in J$^{ASP90}$ using the pTRPE vector to generate the TCRA3V, TCRA11V, and TCRB7R J$^{ASP90}$ reporter cell lines. These cell lines were sorted to yield a unimodal p-HLA multimer+ cell population and expanded for use in functional assays. Sorted TCRA3V, TCRA11V, and TCRB7R J$^{ASP90}$ reporter cell lines were mixed 1:1 with HLA-SCD-expressing K562 cells pulsed with titrated peptide concentrations (10 μM–1 pM). After 16–20 h, cells were analyzed by flow cytometry to determine the percentage of eGFP+ cells in each sample. Cells activated with PMA (50 ng/ml) and Ionomycin (750 ng/ml) were included as positive controls (%GFP$_{max}$), and J$^{ASP90}$ reporter cells cultured in media alone were used as negative controls (%GFP$_{min}$). Data were fitted to a dose-response curve by linear non-regression analysis using GraphPad Prism version 7.0.

**TCRs as probes for recognition of mKRAS epitopes.** Tumor cell lines ($2.5 \times 10^5$ cells/well) were cocultured with TCR transduced J$^{ASP90}$ cells at a 1:1 ratio in 48-well flat bottom plates in a final volume of 500 μl of media (RPMI, 10% FBS, L-Glutamine, Penicillin/Streptomycin). Each tumor cell line was tested in its parental (non-HLA matched) and HLA-engineered (HLA-matched) form using lentiviral particles encoding HLA-SCD constructs of interest and flow cytometric sorting to >99% purity based on the coexpression of the target HLA class I allele and eGFP. G12R+ tumor cell lines were pre-cultured with IFN-γ (500 U/ml) for 48 h prior to co-culture with TCRB7R transduced J$^{ASP90}$ cells. After 16–20 h of incubation at 37 ℃, cells were analyzed by flow cytometry to determine the percentage of live, CD8+mCherry+eGFP+ cells in each sample. Cells activated with PMA and Ionomycin were included as positive controls.

**Expression of mKRAS-specific TCRs in primary CD8+ T cells**

*Primary T cell expansion.* On day 0, human primary CD8+ T cells were activated and expanded using anti-CD3/CD28 antibody–conjugated paramagnetic microbeads (Life Technologies). On day 1 the cells were transduced with lentiviral vector particles containing TCRA3V, TCRA11V, or TCRB7R genes at an MOI of 5. Cell cultures and maintained in media supplemented with IL-7 (5 ng/ml; R&D Systems) and IL-15 (5 ng/ml; R&D Systems). On day 5, T cell cultures were harvested and paramagnetic microbeads were magnetically removed prior to electroporation.

*Cas9 RNP assembly and electroporation.* Successful knock out of the endogenous TCR was carried out using single guide RNAs targeting *TRAC* (5'-TGTGCTA GACATGAGGTCTA-3') and *TRBC1/TRBC2* (5'-GGAGAATGACGAGTGGA CCC-3')[38] obtained from Integrated DNA Technologies. For Cas9 and sgRNA delivery, the ribonucleoprotein (RNP) complex was first pre-formed by incubating TrueCut Cas9 Protein V2 (10 ug, ThermoFisher Scientific) with sgRNA (5 ug) for 10 min at RT. Cells were spun down at $300 \times g$ for 10 min and resuspended at a concentration of $5 \times 10^6$ cells/100 μl in the P3 buffer. Both TCR α/β RNP

complexes and 100 μl of resuspended cells were combined and electroporated (P3 Primary Cell 4-D Kit, Lonza:PBP3-02250). TCRαβ$^{null}$ T cells were then cultured in media supplemented with IL-7 and IL-15 for a total of 14 days.

*Secondary T cell expansion.* T cell cultures were harvested and re-stimulated with peptide-pulsed (500 ng/ml) irradiated (10,000 Rads) HLA-SCD/4-1BBL expressing K562 artificial APCs at a 1:1 ratio. Cultures were supplemented with IL-7 and IL-15 every 48 h until culture termination. Engineered TCR expression was then assessed by CD3, CD8, and TCRαβ expression as well as p-HLA multimer binding then cryopreserved for use in subsequent assays.

**Cytotoxicity assays**

*$^{51}$Cr-release assay.* K562 cells and KRAS-mutated tumor cell lines were labeled with 25 μCi $^{51}$Cr in the presence or absence of peptide (10 μM) for 1 h at 37 °C, washed and tested as targets in a standard 4 h $^{51}$Cr-release assay. Effector cells consisted of primary gene-edited (TCRα-/TCRβ-) mKRAS-specific TCR CD8$^+$ T cells, designated TCRA3V, TCRA11V, TCRB7R. Transgenic TCR expression was assessed by p-HLA multimer assay. Assays were performed, in triplicate, at various effector: target ratios. Data were collected using a MicroBeta2 LumiJET Microplate Counter (PerkinElmer). Data are represented as percent-specific lysis reported as mean ± standard deviation (SD).

*Real-time apoptotic cell death analysis.* Real-time apoptotic cell death analysis (live cell imaging with cellular impedance) was performed to assess extended cytotoxic activity using the xCELLigence Real Time Cell Analysis eSight system (ACEA Biosciences). Target tumor cells were plated ($1 \times 10^4$ cells/well) and allowed to adhere for 24 h. Effector T cells were added at E:T ratios 10:1, 3:1, and 1:1, and the media was supplemented with Annexin V-CF594 (Biotium, Fremont, CA). Cell index (relative cell impedance) was monitored every 15 min for 5 days and normalized to the maximum cell index value immediately prior to effector-cell plating. Shaded lines reflect the mean of duplicate wells ± SD. Concurrent time lapse video monitoring was performed with acquisition of brightfield, green (GFP), and red (CF594) windows every hour.

**CORL23 lung xenograft mouse model**. All animal procedures were performed according to the approved University of Pennsylvania Institutional Animal Care and Use Committee (IACUC) protocol 804226. NSG (NOD/*scid*/γcnull) mice were purchased from the University of Pennsylvania Stem Cell Xenograft Core (SCXC) and housed in micro-isolator cages under sterile conditions in the SCXC AAALAC accredited animal facility at the University of Pennsylvania (temperature: 24 °C, humidity: 50–60%, dark/light cycle: 12 h/12 h). The CORL23 xenograft model was established by intravenous tail injections of $5 \times 10^5$ CORL23-A3 or CORL23-A11 cells expressing CBR luciferase for bioluminescence imaging (BLI). On day 4 post tumor inoculation, mice were treated with $1 \times 10^7$ mKRAS-TCR engineered CD8$^+$ T cells (TCRA3V for CORL23-A3 Cohort, TCRA11V for CORL23-A11 Cohort). Control mice either received no treatment (Mock) or $1 \times 10^7$ TCR KO CD8$^+$ T cells. BLI was used to monitor tumor growth by intraperitoneal injections of 3 mg D-luciferin/150 ul PBS per mouse (~120 mg/kg) and imaged after 10 min using an IVIS Spectrum imaging system. Data were analyzed using Living Image Version 4.5.2 software (PerkinElmer).

**Statistics and reproducibility**. Statistical analysis of multiple comparisons was performed using one-way ANOVA with Tukey's HST post-test, and comparisons between just two groups were performed using Student's unpaired *t*-test. Significance of overall survival was determined via Kaplan–Meier analysis with log-rank (Mantel-Cox) analysis. Data were analyzed using Microsoft Excel (version 16.16.27), and all statistical analyses were performed with Graphpad Prism 7.0 (GraphPad), except analysis of tumor growth and survival curves, which were performed using linear mixed-effects model with Tukey's HSD post-test using the lme4 and the survival package in R. Error bars represent the SD, and $p < 0.05$ was considered statistically significant. *$p < 0.05$, **$p < 0.01$, ***$p < 0.001$, and ****$p < 0.0001$ unless otherwise indicated. ns denotes not significant. All data presented are representative of two or more independent experiments.

**Reporting summary**. Further information on research design is available in the Nature Research Reporting Summary linked to this article.

**Data availability**

MS raw data, DDA and PRM files supporting this study have been deposited in the PRIDE repository [https://www.ebi.ac.uk/pride/archive/projects/PXD024412]. TCRα and TCRβ sequencing data described in this study have been deposited in the NCBI GenBank nucleotide database under the following accession numbers: MZ355910, MZ355911, MZ355912, MZ355913, MZ355914, MZ355915, In Fig. 1b, HLA class I allelic frequencies are supported by data obtained from Allele Frequency Net Database [http://www.allelefrequencies.net] and mKRAS frequencies are supported by data obtained from International Cancer Genome Consortium Data Portal [https://dcc.icgc.org]. KRAS gene expression values reported in Supplementary Table 4 are supported by data obtained from European Bioinformatics Institute (EMBL-EBI) Expression Atlas Portal [https://

www.ebi.ac.uk/gxa/]. The remaining data are available within the article, Supplementary information, or Source Data file. Source data are provided with this paper.

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

## Acknowledgements

We thank the Human Immunology Core at the Univeristy of Pennsylvania for providing leukocytes for research, Drs Nina Luning Prak and Wenzhao Meng for assistance in TCR sequencing, Drs Rico Buchli, Thomas Osterbye, and Anette Stryhn for assistance in biochemical characterization of peptide epitopes, Dr Andrew Rech for bioinformatic analysis, Bereket Gebregziabher and Dr Nune Markosyan for statistical analysis, and Drs Carl June and Elizabeth Jaffee for advice and helpful discussions. This work was supported by NIH R01 CA204261 (to B.M.C. and G.P.L.), NIH P30 CA016520 (to R.H.V.), NIH CA196539 and CA232568 (to B.A.G.), SU2C/Lustgarten Foundation Pancreatic Cancer Collective (to R.H.V. and B.M.C.), the Parker Institute for Cancer Immunotherapy (to B.M.C., G.P.L., and R.H.V.), and Institute for Immunology at University of Pennsylvania (to A.S.B. and G.P.L.). Training grants provided support to A.S.B. (K12 CA076931) and T.B. (T32 CA009140).

## Author contributions

Conceptualization: R.H.V., G.P.L., B.M.C.; methodology: A.S.B., T.B., B.M.C., G.P.L., J.C., M.J.F.; software: L.P.R.; validation: M.L.B., C.X., T.B.; formal analysis: B.A.G., M.J.F., B.M.C.; investigation: A.S.B., T.B., J.C., M.J.F., C.X., M.L.B., S.M.C., C.C., A.D.P., M.H.O.; resources: A.S., D.J.P., A.D.P.; data curation: A.S.B., B.M.C., L.P.R.; writing – original draft: A.S.B., G.P.L., B.M.C.; writing – review and editing: A.S.B., R.H.V., G.P.L., B.M.C.; visualization: A.S.B., J.C., R.H.V., G.P.L., B.M.C.; supervision: G.P.L., B.M.C., R.H.V., B.A.G.; funding: G.P.L., R.H.V., B.M.C., B.A.G.

## Competing interests

R.H.V. reports having received consulting fees or honoraria from Medimmune and Verastem; and research funding from Fibrogen, Janssen, and Lilly. He is an inventor on a licensed patent relating to cancer cellular immunotherapy and cancer vaccines, and receives royalties from Children's Hospital Boston for a licensed research-only monoclonal antibody. A.S.B., R.H.V., G.P.L., and B.M.C. have filed a patent application (PCT/US2020/014988) related to the targeting of KRAS for immunotherapy entitled "Compositions and Methods for Targeting Mutant RAS". A.S., A.D.P., and D.J.P. have filed a patent application (PCT/US2019/026378) that incorporates discoveries related to the NFAT reporter system used in this study entitled "Methods and Compositions Comprising a Viral Vector for Expression of a Transgene and an Effector". M.J.F. is a founder and an employee of MS Bioworks. All other authors have no competing interest to declare.
