## [Peer Review File · Nature Communications]

REVIEWER COMMENTS

Reviewer #1 (Remarks to the Author): with expertise in in silico KRAS neoantigen identification

The study of Bear et al approaches a pool of KRAS mutations that are in the radar of many groups since a couple of decades. KRAS mutations are recurrent driver mutations very frequent in PDAC, CRC and lung cancer. As pointed out by the authors, adoptive T cell therapy targeting these mutations can have a significant impact in treatments against cancer. A off-the-shelf strategy can strongly benefit from a pool of TCRs recognising these mutations. It would allow a personalised medicine without the need to isolate, clone and test TCRs from patients.

The study adopts a multi-omics approach that has been developed in various versions by other groups in the last decade.

The study confirms the presentation of several neoepitopes in cancer cell lines and cell lines transfected with mini-genes. They identify neoepitopes in the HLA-I immunopeptidomes, which is the first evidence of this type, as far as I know.

The authors also confirmed that CD8+ T cells can recognise cancer cell lines expressing the KRAS mutations, in agreement with previous studies.

I have some technical comments that the authors might consider:

1. The MS2 comparison of eluted neoepitopes and synthetic peptides is convincing only in some cases. For example, in Fig. 2d,e,i the MS2 spectra could belong to different sequences.

2. The binding affinity of the target peptides to HLA-I complexes is expressed in log IC50. That is confusing. Is it log10? Because $\log_{10} 3.7 \text{ nM}$ is $10^{E3.7} \text{ nM}$, right? Which is a quite low binding affinity.

Why do the authors report the predicted binding affinity as IC50 and the measured binding affinity as log(IC50)?

The authors shall also compare their results of binding affinity with those obtained by other groups. For example, the binding affinity of the neoepitope candidate KRAS5-14V to HLA-A0201 has been recently published by Mishto et al., *Front. Immunol.* 2019, where also its crystal structure has been investigated.

My main concern, however, is related to the novelty of the study. The authors claim that "this study validates mKRAS peptides as bona fide epitopes facilitating the development of immune therapies ..." and "these findings OPEN the path toward rational development of off-the-shelf mKRAS-targeted immunotherapies including vaccine, antibody and adoptive cell therapies". Some of the mKRAS epitopes that the authors claimed to have identified in their study have been already identified by Kubuschok et al (*CCR* 2006) and Wang et al (*CIR* 2016). The latter study applied a different multi-omics approach and identified several mKRAS G12V and G12D neoepitopes. They isolated TCRs and treated xenograft mouse models demonstrating the adoptive T cell therapy targeting those neoepitopes can successfully tackle cancer. Yang lab is carrying out trials on cancer patients using these TCRs.

In addition, Tran et al (*NEJM* 2016) demonstrated that CRC patients with metastasis can successfully be treated with adoptive T cell therapy using TCRs recognising mKRAS G12D mutations.

On top of that, all neoepitopes described in this study are different variants of the same backbone peptide, carrying different KRAS mutations. Also the HLA-I alleles identified as those presenting the neoepitopes were already known and could be easily selected by using common HLA-I-peptide predictors.

Considering that Bear et al did not perform *in vivo* experiments and no treatment in patients, I think that the manuscript shall fit better for a more specialised journal.

Reviewer #2 (Remarks to the Author): with expertise in identification of MHC-restricted tumor antigens

This manuscript is a true tour de force, presenting a comprehensive identification and validation of

several MHC-I restricted epitopes of mutant forms of the KRAS protein. At the translational/clinical level, the work sets the stage to employ these epitopes in vaccination and adoptive T cell transfer immunotherapy approaches. Given the long and largely unsuccessful attempts to target this molecule immunologically, it is difficult to overestimate the importance of the work. In addition, given the frequency with which recurrent KRAS mutations arise in epithelial tumors, these epitopes are substantially more attractive for development than patient specific mutated neoantigens.

At a more fundamental level, the use of predictive methodology, followed by mass spectrometric validation, in vitro immunogenicity, and target expression using cloned T cell receptors illuminates the importance of such a comprehensive approach while simultaneously revealing some of the limitations of each method applied alone. My only significant comment regarding revision of the manuscript is that this last point could be developed and explained much more thoroughly in both the results and the discussion, in conjunction with editing some of the figures for consistency and ease of tracking information.

1. It is very difficult to understand the ultimate value of predicted and measured affinities and complex stability to the processed and presented peptides identified in Figure 2i. It would seem that some epitopes are presented despite being predicted not to be, and the converse, and it would be useful to at least point this out in results and discussion – is there ultimately a “most valuable” algorithm to predict presentation?
2. Similarly, understanding that many peptides are presented based on Fig 2i, they are seemingly not immunogenic in Figure 3. Do the authors have any insight into this? Can it be related to differences in binding affinity and/or complex stability, since these would be more relevant in pulsing APC with synthetic peptide?
3. It is very difficult to keep track of the various mutant peptide forms, and their MHC restriction elements, that are characterized in Fig 1b-h; Table 1; Figure 2; Figure 3a, c, e; and Figure S1. Some of this is a nomenclature issue, in that the peptides are identified in multiple ways. Also, some figures include more peptides (Fig 1b, 2i) than others (Fig 1), and not all of these are apparently high frequency KRAS mutations (A, S). At the least, perhaps these could be relegated to the right side so that the C-D-R-V presentation order is maintained for consistency.
4. In some instances, the MHC restriction element is omitted from the figure itself (Fig 3a, c, e), necessitating referral to the legend or the text.
5. Moving from Fig 1 / Table 1 to Fig S1 repeatedly to visualize the HLA-A2 data is unnecessary, and I would suggest that the A2 data be merged into Fig 1. Correspondingly, I would suggest that all of the mass spectrometry spectra be placed in supplemental figures – while important for ms aficionados, they contribute little to the main issues in the manuscript.
6. The authors comment on the lack of detection of cysteine mutated KRAS peptides, despite the fact that they are immunogenic, and make reference to the possibility that the cysteine might be oxidized. This is often observed, and could have been easily detected by mass spectrometry. A more subtle modification with known immunological relevance (Meadows et al *Immunity* 6:273 (1997); Kittlesen et al, *J Immunol* 160: 2099 (1998)) is the formation of cystine through disulfide bonding to free cysteine, which is common in serum and plasma. Have the authors looked for either modification directly?

Victor Engelhard

Reviewer #3 (Remarks to the Author): with expertise in computational prediction of Neoepitopes

Bear et al. report an evaluation of KRAS mutations for T cell targeting by discovering epitopes presented by common HLA. They used genomics based prediction followed by biochemical assays to identify which mutant KRAS (mKRAS) peptides would bind and be presented by HLA class I alleles. They then discovered specific T cell receptor alpha/beta sequences that bind mKRAS peptides presented by HLA (using a really nice reporter system) and elicit antigen-specific cytotoxic activity when expressed by TCR alpha/beta (null) primary CD8+ T cells. T cell mediated killing of a tumor cell line engineered to express specific monoallelic HLAs was proportional to the abundance of peptide/MHC protein complexes on the target cell surface. In my view this is highly important work. KRAS is a prevalent oncogene which is difficult to target clinically using small molecule inhibitors, however the conserved RAS G12D mutation has been successfully targeted by

adoptive transfer of neoantigen-specific tumor infiltrating lymphocytes (Tran et. al 2016 NEJM) and is predicted to elicit the most common neoantigen across The Cancer Genome Atlas dataset (Thorsson et al. Immunity 2018).

This is an exceptionally well done study with clear logic and rigorous experimentation. Many studies of neoantigens in cancer stop at the stage of genomics based prediction, without proceeding to biological validation by proteomics and/or T cell functional studies. It is excellent that the neoantigens reported in this study have been carefully and robustly validated by binding assays, T cell culture, and anti-tumor cytotoxicity associated with antigen-specific TCR engagement (which in the end, is a key variable of interest for CD8+ T cells).

This manuscript was honestly a breath of fresh air to read. I look forward to this being published so I can use it as a Journal Club article for my lab.

Comments:

1. Lines 306-308: It is not necessarily true that the graded pattern of NFAT activation in the reporter cell co-culture experiment implies variable cell surface expression of peptide/HLA here. This phenotype could also be explained, for example, by different coreceptor expression patterns (which may also be altered by interferon-gamma treatment).
2. Line 352-353: Why is "data not shown" for the lack of endogenous HLA class I for the COR-L3? Can this be added as a supplemental figure panel?

Reviewer #4 (Remarks to the Author): with expertise in identification and validation of novel tumor-associated antigens

The authors investigate the relevance of KRAS G12 mutations in the context of the development of novel immunotherapies targeting mutated peptide ligands derived from these mutations. KRAS mutations are known as driver mutations in many malignancies and the characterization of the G12 mutations resulting in potential epitopes on tumor cells which can be recognized by the immune system is highly relevant and may lead to a more generalized approach in a significant number of patients. The authors apply a number of state of the art (although basically previously published) methods to investigate this topic.

Overall the results shown here are interesting and suggest presentation of KRAS G12 mutations as peptide ligands in the context of HLA-A*0301, HLA-A*1101 and HLA-B*0702. However, significant controls are missing in order to fully convince the readership that mutated KRAS G12 peptides are sufficiently naturally presented and clinically relevant T-cell responses could be detected here. Specificity and cross reactivity of the identified TCR have not fully been demonstrated and sensing of HLA/mutKRASG12 on tumor cells as claimed by the authors is not definitively proven by the experiments shown here.

Major aspects:

- To better understand the quality of MS data, a table with complete primary MS information including all scores should be provided also including the whole peptidome found by the analysis.
- The authors use a rather artificial system of peptide presentation in experiments shown in Figure 1 and 6. MS data derived from patient tumor samples or tumor cell lines expressing defined HLA alleles and harboring the respective mutations are missing. The following questions are highly relevant for clinical translation:
 - o How do the defined mutated peptides resulting from natural gene expression compete with other peptides in HLA presentation for these and other HLA molecules. The authors need to compare their MS data derived from these cell lines with tumor samples or at least cell lines having defined mutations and HLA alleles. Did the authors look for cell lines naturally expressing relevant HLA alleles and defined KRAS G12 mutations? Otherwise, mutations could be introduced in selected cell lines.
 - o What is the impact of allele frequency and heterogeneity within the tumor on peptide presentation on tumor cells? Again, the authors need to compare there MS data with patient-derived tumor material having defined mutations.

- o An experiment is missing demonstrating specific recognition of naturally presented peptides on target cells. K562 cells need to be included which are not only peptide-pulsed (Fig. 4b) but also transduced by TMC incorporating mutated compared to wildtype sequences to demonstrate this. Although TMC-expressing K562 cells have been used for the 51Cr assay and were recognized by TCR-transduced cells as shown in Fig. 5b, a control using TMC wildtype-transduced K562 is missing here. In addition, tumor cell lines without the respective mutations should be tested for recognition (Fig 4c-e and Fig. 5c and d).
- o Potential cross reactivity of the TCR should be investigated by alternative peptide scans in order to determine the specificity of the TCR and excluding major cross reactivity potentially involved in recognition of targets used in these experiments.
- o Is the tumor reactivity shown by these TCR strong enough for tumor eradication? Animal models using xenogeneic mouse models to test tumor control could be useful to test this.
- o Well described control peptides should be included for correlation of quantification of cell surface expression shown in Fig. 6e and f. Moreover, the analysis of quantitative expression in such an artificial system is of questionable relevance. Again, a table should be included showing all peptides identified as well as important quality control information.

Additional aspects:

- Figure 1c-h and S1b and c: Were positive and negative controls included in all experiments? In contrast to Figure 1 c-h, for S1b and S1c it is unclear if only gp100 or others controls have been used. It is not indicated how many times that experiment has been repeated.
- Figure S2d and e: The control should be defined in the legend (isotype?) and K562 Mock-transduced cells should be shown.
- Fig. 3B, d and f: The negative control for multimer analyses needs to be defined. At least two controls are needed: 1. T cells stimulated with alternative peptides and 2. wildtype multimers.
- Cytotoxicity of these TCR requires high E:T ratios as shown in Fig. 5 and Fig. 6. Again is this sufficient for tumor rejection?
- The discussion is poorly addressing limitations of the study and potential challenges of further clinical translation.
- Significant publications in the field should be cited (e.g. 10.1172/JCI99538).
- The manuscript needs some more editing.

To Reviewers:

We wish to thank each reviewer for their thoughtful critique and helpful suggestions to improve our manuscript. We spent considerable time and effort addressing each comment and hope that our response adequately addresses each point.

Reviewer 1

The study of Bear et al approaches a pool of KRAS mutations that are in the radar of many groups since a couple of decades. KRAS mutations are recurrent driver mutations very frequent in PDAC, CRC and lung cancer. As pointed out by the authors, adoptive T cell therapy targeting these mutations can have a significant impact in treatments against cancer. A off-the-shelf strategy can strongly benefit from a pool of TCRs recognising these mutations. It would allow a personalised medicine without the need to isolate, clone and test TCRs from patients. The study adopts a multi-omics approach that has been developed in various versions by other groups in the last decade.

The study confirms the presentation of several neoepitopes in cancer cell lines and cell lines transfected with mini-genes. They identify neoepitopes in the HLA-I immunopeptidomes, which is the first evidence of this type, as far as I know.

The authors also confirmed that CD8+ T cells can recognise cancer cell lines expressing the KRAS mutations, in agreement with previous studies.

Response: Thank you for the thoughtful comments.

I have some technical comments that the authors might consider:

1. The MS2 comparison of eluted neoepitopes and synthetic peptides is convincing only in some cases. For example, in Fig. 2d,e,i the MS2 spectra could belong to different sequences.

Response: Thank you for bringing this to our attention. For some samples, eluted peptide was orders of magnitude less abundant than the corresponding synthetic peptide. Although the most abundant ions match, background and interference may have affected spectra interpretation. So, to further support the MS spectra data, we have now provided isotopic dot product (idotp) and dot product (dotp) values as generated in the targeted proteomic analysis using Skyline (MacLean et al, Ref. 55). Both values provide a measurement of similarity between eluted and expected or reference peptide. Idotp provides a measure to assess precursor isotope distribution and its correlation between the observed (eluted) and expected (synthetic) peptide while dotp provides a measure of similarity between the observed and library spectrum (Proteome Discovery Database) peak areas. Values range from its best of 1.0 to its worst of 0.0. As now shown in new Figs S5-7, all precursor ions have idotp values >0.90 and product ions with dotp values >0.50 supporting our assignment of identity between eluted and synthetic peptides. We hope this additional information provides confidence in the identification of these less abundant peptides.

2. The binding affinity of the target peptides to HLA-I complexes is expressed in log IC50. That is confusing. Is it log10? Because log10= 3.7 nM is 10E3.7 nM, right? Which is a quite low binding affinity.

Why do the authors report the predicted binding affinity as IC50 and the measured binding affinity as log(IC50)?

Response: Thank you for this suggestion. The peptide binding affinity values reported in our study are derived from competitive binding assays originally developed in the Hildebrand laboratory

(Buchli et al, Ref. 30), and we have now provided as Fig S2 representative results for selected peptide-HLA class I pairs to provide the audience with a better understanding of how these values are derived. To allow comparison between experimental and predicted values, all affinity determinations were transformed logarithmically and reported as such in Table S1. Additionally, to determine whether experimental and predicted values are in general agreement, we have plotted values against each other in a log-log format (new Fig S3). Our results demonstrate a strong linear correlation ($R^2 > 0.9106$) between experimental and predicted affinities of 9-mers (8-16C-V) and 10-mers (7-16C-V) binding to HLA-A*03:01 and HLA-A*11:01. In contrast, experimental determinations of mKRAS peptide binding to HLA-A*02:01 and HLA-B*07:02 indicate that most of these peptides display much lower affinities than predicted leading to a poor correlation between these 2 values ($R^2 = 0.1033$ and $R^2 = 0.6945$, respectively). Interestingly, with the exception of mKRAS 10-19R and 10-19D in the context of HLA-B*07:02, none of the A2 or B7 predicted peptides could be identified in the mass spectrometry analysis. Thus, even for high frequency alleles such as HLA-A*02:01, experimental affinity and stability determinations are valuable tools for epitope identification; however, all predictors have limitations and may not always correlate with presentation on HLA class I molecules.

The authors shall also compare their results of binding affinity with those obtained by other groups. For example, the binding affinity of the neoepitope candidate KRAS5-14V to HLA-A0201 has been recently published by Mishto et al., *Front. Immunol.* 2019, where also its crystal structure has been investigated.

Response: The candidate peptide reported by Mishto et al. (Ref. 45) is a spliced peptide (KL_VVGAVGV) lacking V at position 7 of the canonical peptide. This spliced peptide was generated from a longer 2-35V peptide via in vitro proteasome digestion followed by MS measurements. The authors could not identify the linear 5-14V/HLA-A*02:01-restricted peptide in their study consistent with our MS data (now mentioned on lines 393-395). Thank you again for bringing this to our attention! The emerging field of spliced peptides is beyond the scope of the current study. A review of the literature did not yield any other studies reporting on affinity measurements for mKRAS peptides and thus, the experimental affinity data reported in our manuscript fills an important void in the literature.

Additional Comments from Reviewer #1:

My main concern, however, is related to the novelty of the study. The authors claim that “this study validates mKRAS peptides as bona fide epitopes facilitating the development of immune therapies ...” and “these findings OPEN the path toward rational development of off-the-shelf mKRAS-targeted immunotherapies including vaccine, antibody and adoptive cell therapies”. Some of the mKRAS epitopes that the authors claimed to have identified in their study have been already identified by Kubuschok et al (CCR 2006) and Wang et al (CIR 2016). The latter study applied a different multi-omics approach and identified several mKRAS G12V and G12D neoepitopes. They isolated TCRs and treated xenografts mouse models demonstrating the adoptive T cell therapy targeting those neoepitopes can successfully tackle cancer. Yang lab is carrying out trials on cancer patients using these TCRs.

In addition, Tran et al (NEJM 2016) demonstrated that CRC patients with metastasis can successfully be treated with adoptive T cell therapy using TCRs recognising mKRAS G12D mutations.

On top of that, all neoepitopes described in this study are different variants of the same backbone peptide, carrying different KRAS mutations. Also the HLA-I alleles identified as those presenting

the neoepitopes were already known and could be easily selected by using common HLA-I-peptide predictors.

Response: We certainly agree that other investigators have made important contributions regarding mKRAS epitope identification and have acknowledged each contribution in our manuscript. Thank you for bringing the Kubuschok paper to our attention. Kubuschok et al (Ref. 46, PMID: 16489095) reported ELISPOT and lysis assays obtained from patient CD4+ and CD8+ T cells after stimulation with peptides. This study did not biochemically characterize or proteomically validate individual HLA class I epitopes with binding assays or MS, nor did it confirm T cell specificity by pMHC tetramer assays or isolation of $\alpha\beta$ TCR heterodimers. In our manuscript, we cited Wang et al (Ref. 21, PMID: 26701267) as evidence that mKRAS-specific TCRs can be isolated from immunized HLA-A*11:01 transgenic mice. We also cited the study by Tran et al (Ref. 22, PMID: 27959684) which highlights the potential for T cell immunotherapy against a HLA-C*08:02 restricted G12D epitope. These studies plus others provided the inspiration for our work related to mKRAS epitope identification and validation.

Regarding comment on “variants of the same backbone”: G12 variants are present in over 75% of KRAS mutated cancers with relative frequencies of 94%, 91% and 76% in PDAC, LAC and CRC, respectively (PMID: 25323927). The high frequency of these G12 variants in cancer provides a rationale for why our study has centered on epitopes derived from AA substitutions at the most frequent hot spot (codon 12) position. We have now included a sentence in the introduction to strengthen this point (line 108-110). Our manuscript provides strong biochemical evidence for several new epitopes, including 7-16D/A*03:01, 8-16V/A*03:01, 7-16V/A*03:01, 8-16R/A*03:01, 10-19D/B*07:02 and 10-19R/B*07:02 (new Figs S5-7). In addition, $\alpha\beta$ TCR transfer in CRISPR-edited normal donor CD8+ T cells as well as Jurkat T cells (J^{ASP90}) provides unambiguous and clear evidence of HLA-I restricted recognition of 7-16V/A*03:01, 7-16VA*11:01, and 10-19R/B*07:02 mKRAS on a variety of cell lines.

Considering that Bear et al did not perform and in vivo experiments and no treatment in patients, I think that the manuscript shall fit better for a more specialized journal.

Response: We have now included adoptive transfer of TCR-expressing primary edited CD8+ T cells into tumor-bearing NSG mice (new Fig 6). Tumor regression as measured by bioluminescence and improved overall survival are shown in tumor bearing animals after adoptive transfer of mKRAS-specific TCRs compared to control groups.

Reviewer 2

This manuscript is a true tour de force, presenting a comprehensive identification and validation of several MHC-I restricted epitopes of mutant forms of the KRAS protein. At the translational/clinical level, the work sets the stage to employ these epitopes in vaccination and adoptive T cell transfer immunotherapy approaches. Given the long and largely unsuccessful attempts to target this molecule immunologically, it is difficult to overestimate the importance of the work. In addition, given the frequency with which recurrent KRAS mutations arise in epithelial tumors, these epitopes are substantially more attractive for development than patient specific mutated neoantigens.

At a more fundamental level, the use of predictive methodology, followed by mass spectrometric validation, in vitro immunogenicity, and target expression using cloned T cell receptors illuminates the importance of such a comprehensive approach while simultaneously revealing some of the limitations of each method applied alone. My only significant comment regarding revision of the manuscript is that this last point could be developed and explained much more thoroughly in both the results and the discussion, in conjunction with editing some of the figures for consistency and ease of tracking information.

Response: We very much appreciate the thoughtful comments. We have now revised the manuscript figures and nomenclature to facilitate reading of manuscript.

1. It is very difficult to understand the ultimate value of predicted and measured affinities and complex stability to the processed and presented peptides identified in Figure 2i. It would seem that some epitopes are presented despite being predicted not to be, and the converse, and it would be useful to at least point this out in results and discussion – is there ultimately a “most valuable” algorithm to predict presentation?

Response: An important issue has been raised regarding the value of predictive algorithms to identify antigen epitopes. Most would agree that prediction algorithms based on peptide affinity and stability have general utility to guide scientists toward selecting candidate; however, over-reliance on prediction algorithms is fraught with pitfalls as suggested by data in new Fig S3. More comprehensive approaches with biochemical assays and MS analysis combined with functional assays using cloned TCRs is essential to ensure accuracy.

2. Similarly, understanding that many peptides are presented based on Fig 2i, they are seemingly not immunogenic in Figure 3. Do the authors have any insight into this? Can it be related to differences in binding affinity and/or complex stability, since these would be more relevant in pulsing APC with synthetic peptide?

Response: Thank you for raising this issue. We do not have any new insights related to peptide immunogenicity. Although our sample size is small (n=4 donors) for most mKRAS peptides, our data is fairly consistent. For example, 4/4 donors elicit measurable CD8 T cells to G12V/A*11:01 and G12C/A*11:01, while no donor responded to G12R/A*11:01 with in vitro assays using mature autologous peptide-pulsed DC (new Fig 2A). We were disappointed to find that all G12x peptides restricted to HLA-A*02:01 were non-immunogenic (new Fig 2A) despite several older reports in the literature which claim the induction of G12D or G12V/HLA-A*02:01 restricted CD8+ T cells in patients (Abrams et al, Ref. 17; Kubuschok et al, Ref. 46). In depth characterization of vaccine-induced T cells will be critical toward resolving the question of A*02:01 (and sub-alleles) -restricted mKRAS epitopes in human cancer.

3. It is very difficult to keep track of the various mutant peptide forms, and their MHC restriction elements, that are characterized in Fig 1b-h; Table 1; Figure 2; Figure 3a, c, e; and Figure S1. Some of this is a nomenclature issue, in that the peptides are identified in multiple ways. Also, some figures include more peptides (Fig 1b, 2i) than others (Fig 1), and not all of these are apparently high frequency KRAS mutations (A, S). At the least, perhaps these could be relegated to the right side so that the C-D-R-V presentation order is maintained for consistency.

Response: Thank you for bringing this matter to our attention. We agree and have attempted to simplify the designation of mKRAS peptides. We have also removed data pertaining to the low frequency mKRAS alleles- G12A and G12S since further immunological studies were not performed.

4. In some instances, the MHC restriction element is omitted from the figure itself (Fig 3a, c, e), necessitating referral to the legend or the text.

Response: Thank you for bringing this to our attention. We have made the appropriate corrections.

5. Moving from Fig 1 / Table 1 to Fig S1 repeatedly to visualize the HLA-A2 data is unnecessary, and I would suggest that the A2 data be merged into Fig 1. Correspondingly, I would suggest that all of the mass spectrometry spectra be placed in supplemental figures – while important for ms aficionados, they contribute little to the main issues in the manuscript.

Response: Thank you for suggesting this change and we have complied by moving the MS data into the supplemental portion of the manuscript. In addition, the HLA-A*02:01 data has been relocated back to the main manuscript figures.

6. The authors comment on the lack of detection of cysteine mutated KRAS peptides, despite the fact that they are immunogenic, and make reference to the possibility that the cysteine might be oxidized. This is often observed, and could have been easily detected by mass spectrometry. A more subtle modification with known immunological relevance (Meadows et al Immunity 6:273 (1997); Kittlesen et al, J Immunol 160: 2099 (1998)) is the formation of cystine through disulfide bonding to free cysteine, which is common in serum and plasma. Have the authors looked for either modification directly?

Response: We concur with this suggestion but unfortunately, our MS analysis did not examine either modification. We have inserted a brief comment in the manuscript (line 234) for clarification.

Reviewer 3

Bear et al. report an evaluation of KRAS mutations for T cell targeting by discovering epitopes presented by common HLA. They used genomics based prediction followed by biochemical assays to identify which mutant KRAS (mKRAS) peptides would bind and be presented by HLA class I alleles. They then discovered specific T cell receptor alpha/beta sequences that bind mKRAS peptides presented by HLA (using a really nice reporter system) and elicit antigen-specific cytotoxic activity when expressed by TCR alpha/beta (null) primary CD8+ T cells. T cell mediated killing of a tumor cell line engineered to express specific monoallelic HLAs was proportional to the abundance of peptide/MHC protein complexes on the target cell surface. In my view this is highly important work. KRAS is a prevalent oncogene which is difficult to target clinically using small molecule inhibitors, however the conserved RAS G12D mutation has been successfully targeted by adoptive transfer of neoantigen-specific tumor infiltrating lymphocytes (Tran et. al 2016 NEJM) and is predicted to elicit the most common neoantigen across The Cancer Genome Atlas dataset (Thorsson et al. Immunity 2018).

This is an exceptionally well done study with clear logic and rigorous experimentation. Many studies of neoantigens in cancer stop at the stage of genomics based prediction, without proceeding to biological validation by proteomics and/or T cell functional studies. It is excellent that the neoantigens reported in this study have been carefully and robustly validated by binding assays, T cell culture, and anti-tumor cytotoxicity associated with antigen-specific TCR engagement (which in the end, is a key variable of interest for CD8+ T cells).

This manuscript was honestly a breath of fresh air to read. I look forward to this being published so I can use it as a Journal Club article for my lab.

Response: **We very much appreciate the thoughtful comments.**

1. Lines 306-308: It is not necessarily true that the graded pattern of NFAT activation in the reporter cell coculture experiment implies variable cell surface expression of peptide/HLA here. This phenotype could also be explained, for example, by different coreceptor expression patterns (which may also be altered by interferon-gamma treatment).

Response: **Thank you for offering this suggestion. We concur and have modified our conclusions to incorporate this interpretation (lines 304-306).**

2. Line 352-353: Why is “data not shown” for the lack of endogenous HLA class I for the COR-L3? Can this be added as a supplemental figure panel?

Response: **Thank you for pointing this out. Upon review, this statement was made in error. We have now included the HLA class I expression for COR-L23 in new Fig S11 as suggested. We have also provided rationale for selecting cell line COR-L23 given its ability to recapitulate a model of metastatic lung cancer in NSG mice.**

Reviewer 4

The authors investigate the relevance of KRAS G12 mutations in the context of the development of novel immunotherapies targeting mutated peptide ligands derived from these mutations. KRAS mutations are known as driver mutations in many malignancies and the characterization of the G12 mutations resulting in potential epitopes on tumor cells which can be recognized by the immune system is highly relevant and may lead to a more generalized approach in a significant number of patients. The authors apply a number of state of the art (although basically previously published) methods to investigate this topic.

Overall the results shown here are interesting and suggest presentation of KRAS G12 mutations as peptide ligands in the context of HLA-A*0301, HLA-A*1101 and HLA-B*0702. However, significant controls are missing in order to fully convince the readership that mutated KRAS G12 peptides are sufficiently naturally presented and clinically relevant T-cell responses could be detected here. Specificity and cross reactivity of the identified TCR have not fully been demonstrated and sensing of HLA/mutKRASG12 on tumor cells as claimed by the authors is not definitively proven by the experiments shown here.

Response: **Thank you for the suggestions. We have carefully reviewed each figure and have made the appropriate changes to include the most appropriate controls.**

Major aspects:

1. To better understand the quality of MS data, a table with complete primary MS information including all scores should be provided also including the whole peptidome found by the analysis.

Response: **Please refer to new supplementary Files 2 and 3. Please also refer to response to Reviewer #1 / Comment 1 for additional information on MS/MS data interpretation.**

2. The authors use a rather artificial system of peptide presentation in experiments shown in Figure 1 and 6.

Response: The use of monoallelic lines for characterization of HLA peptidomes is a workflow used in many laboratories today and is supported by numerous high-quality publications (PMID: 28228285, PMID: 31495665, PMID: 9089095, PMID: 31527070, PMID: 33298915). Please note that quantitation of p-MHC complexes (originally shown in Fig 6 and now relocated to new Fig S13 in the revised manuscript) was performed using the COR-L23 lung carcinoma cell line which encodes the endogenous G12V mKRAS allele and engineered to express either A*03:01 or A*11:01 at physiological levels (see new Fig S12). Thus, we hope this experiment satisfies the reviewers request for additional information regarding MS data in tumor cell lines.

MS data derived from patient tumor samples or tumor cell lines expressing defined HLA alleles and harboring the respective mutations are missing. The following questions are highly relevant for clinical translation:

How do the defined mutated peptides resulting from natural gene expression compete with other peptides in HLA presentation for these and other HLA molecules. The authors need to compare their MS data derived from these cell lines with tumor samples or at least cell lines having defined mutations and HLA alleles. Did the authors look for cell lines naturally expressing relevant HLA alleles and defined KRAS G12 mutations? Otherwise, mutations could be introduced in selected cell lines.

Response: Our team has placed significant effort and resources on assembling a panel of 74 cell lines and organoids (see Table attached to this letter). However, with the exception of 4 cell lines matched for mKRAS G12V/ HLA-A*03:01, we have been unable to identify cell lines that matched for either mKRAS G12V/HLA-A*11:01 or mKRAS G12R/HLA-B*07:02. The one cell line (SNG-M) reported to be G12V /HLA-A*11:01, lacks HLA class I expression due to bi-allelic β 2m frameshift mutation. We have now expanded the functional studies with one of mKRAS G12V / HLA-A*03:01 matched cell line (NCI-H441) and these results are shown in new Fig 5. To perform quantitative proteomics and xenograft model, we selected the COR-L23 cell line for HLA class I transfection while preserving natural mKRAS expression. Of significance, COR-L23 cell line has been recently reported to recapitulate a model of metastatic lung cancer (Jin et al. Ref. 41). As shown in Fig S12, the HLA class I expression levels on engineered COR-L23 lines is physiologic as assessed by flow cytometry. Moreover, the number of p/HLA-I complexes quantitated on engineered COR-L23 lines (new Fig 5) is similar to that reported for other mKRAS complexes (Wang et al., Ref. 36) in non-engineered tumor (HS758.T) cell lines and engineered COS-7 cells (40 to 196 p/HLA-I complexes per cell). Similarly, our MS approach identified both the 9-mer and 10-mer mKRAS G12V epitope bound to HLA-A*03:01 as initially reported for G12D mKRAS by Wang et al (Ref. 36). The suggestion to introduce a KRAS mutation in WT cells to create isogenic cell line is appreciated and we will certainly consider this approach in the future. Finally, the challenge of finding matched p/HLA-I cell lines is one that many investigators have also encountered (PMID: 30683863, PMID:31527070, PMID: 26701267), so we hope each reviewer appreciates our efforts and the significance of the data presented in our manuscript.

What is the impact of allele frequency and heterogeneity within the tumor on peptide presentation on tumor cells? Again, the authors need to compare there MS data with patient-derived tumor material having defined mutations.

Response: Although we agree this is an interesting question, it is beyond the scope of our study. Direct detection of peptide epitopes from clinical material is a complex and arduous task and will be the topic of a future study.

An experiment is missing demonstrating specific recognition of naturally presented peptides on target cells. K562 cells need to be included which are not only peptide-pulsed (Fig. 4b) but also transduced by TMC incorporating mutated compared to wildtype sequences to demonstrate this. Although TMC-expressing K562 cells have been used for the 51Cr assay and were recognized by TCR-transduced cells as shown in Fig. 5b, a control using TMC wildtype-transduced K562 is missing here. In addition, tumor cell lines without the respective mutations should be tested for recognition (Fig 4c-e and Fig. 5c and d).

Response: To validate recognition of 7-16V in the context of HLA-A*03:01 and HLA -A*11:01 and 10-19R in the context of HLA-B*07:02, two set of experiments are presented. First, J^{ASP90} reporter cells engineered to express individual $\alpha\beta$ TCRs were tested against a panel of G12 peptide variants (new Fig 3b). This data clearly established that TCRs directed at 7-16V/A*03:01 and 10-19R/B*07:02 do not cross-react with any of the other G12 variants. We also showed that the TCR directed at 7-16V/A*11-01 not only recognize 7-16V but also cross-reacts with 7-16C. In addition, a no peptide control is included for each group (indicated as 0 on peptide scale on new Fig 3b). None of these TCR show cross-reactivity with wild-type (WT) KRAS peptides at high concentrations (10ug/mL) (new Fig 3b). Given these results, only a K562 cell line expressing TMC that included G12 variants was tested (new Fig 4b) with the goal to demonstrate that mutated epitope is processed and presented (as reported by proteomics in Figs S5-7).

Potential cross-reactivity of the TCR should be investigated by alternative peptide scans in order to determine the specificity of the TCR and excluding major cross-reactivity potentially involved in recognition of targets used in the experiments.

Response: Thank you for the suggestion but we should have been clearer regarding the primary objective of our study which was to focus on epitope discovery and validation. We employed each TCR as a reagent to probe and detect mKRAS peptide/HLA complexes on human tumor cells. Assessment of individual TCR attributes such as potential cross-reactivity to non-cognate peptides is beyond the scope of our report. As we isolate and characterize additional TCRs, we plan to investigate this issue as suggested but for now, this exercise will require significant resources and, in our opinion, will not alter the conclusions of our study.

Is the tumor reactivity shown by these TCR strong enough for tumor eradication? Animal models using xenogeneic mouse models to test tumor control could be useful to test this.

Response: Thank you for offering this suggestion. We have now demonstrated that CRISPR-Cas9 edited primary human CD8+ T cells engineered to express TCRs directed at 7-16V/A*03:01 and 7-16V/A*11:01 promotes tumor rejection and increased survival in a COR-L23 bearing NSG model. This data shown in new Fig 6 confirms the ability of mKRAS-specific TCR transduced primary CD8+ T cells to “sense” less than 100 p-HLA-I per cell (new Fig 5a). We have now included in the discussion section (line 418) our interpretation of this data in the context of number of complexes quantitated presented on the cell surface of cancer cell lines.

Well described control peptides should be included for correlation of quantification of cell surface expression shown in Fig. 6e and f. Moreover, the analysis of quantitative expression in such an

artificial system is of questionable relevance. Again, a table should be included showing all peptides identified as well as important quality control information.

Standard labeled peptides were the controls to derive AUC based on peptide concentration (new Fig S11). Relative to these values, AUC for eluted peptide provide their concentrations which along with cell concentration allows calculation of copy number/cell. Similar methodology has been used by others (Wang et al., Ref 36). The primary proteomic data of CORL23-A3 and CORL23-A11 is now provided in Supplementary File 3.

Additional aspects:

Figure 1c-h and S1b and c: Were positive and negative controls included in all experiments? In contrast to Figure 1 c-h, for S1b and S1c it is unclear if only gp100 or others controls have been used. It is not indicated how many times that experiment has been repeated.

Response: Fig 1c and d have been updated to include positive control peptides (listed in Table S3) as positive controls. Negative controls were included but are not shown (for clarity). Each experiment was performed twice.

Figure S2d and e: The control should be defined in the legend (isotype?) and K562 Mock-transduced cells should be shown.

Response: This data is shown in new Fig S4d and S4e. The K562 parental (mock) cells stained with W6/32 (S4d) and allele-specific mAb (S4e) is shown as the control. The isotype-controls for each cell line are not shown for clarity but are similar (near identical) to the K562 parental cells. A legend has now also been added to the figure for clarity.

Fig. 3B, d and f: The negative control for multimer analyses needs to be defined. At least two controls are needed: 1. T cells stimulated with alternative peptides and 2. wildtype multimers.

Response: This data is new Fig 2. We employed various control p-MHC multimers and T cell sources during our studies to ensure specificity, including a CD8+ T cells activated in the absence of peptide (non-specific control). The accurate representation of p-MHC multimer staining is further validated by the fact that CD8+Multimer+ T cells were flow cytometrically sorted to allow for the isolation of mKRAS-TCR sequences which we go on to further validate in our study.

Cytotoxicity of these TCR requires high E:T ratios as shown in Fig. 5 and Fig. 6. Again, is this sufficient for tumor rejection?

Response: Yes, the new adoptive transfer data shown in new Fig 6 provides evidence that TCR-transduced primary human CD8 T cells can promote in vivo tumor rejection in NSG mice.

The discussion is poorly addressing limitations of the study and potential challenges of further clinical translation. Significant publications in the field should be cited (e.g. 10.1172/JCI99538). The manuscript needs some more editing.

Response: Our revised discussion addresses the study limitations by stating that our work provides a "curated albeit partial set of mKRAS peptides that represent candidate epitopes". The need to further investigate HLA-A*02:01 restricted mKRAS epitopes and reconcile the

observations made by various investigators is required in order to move the field forward. We also discussed the limitations of bioinformatic algorithms to identify candidate epitopes and stress the need for more comprehensive omics-based approaches. Finally, a major challenge for translation - implicit in all personalized medicine-based cell therapies - is the need to screen and identify potential patients in a clinically relevant time frame.

TABLE- Cell Lines and Organoids								
ID	Histology	KRAS	HLA-A1	HLA-A2	HLA-B1	HLA-B2	HLA-C1	HLA-C2
CaLu-1	lung	G12C	26:01	29:02	44:03	15:01	03:04	16:01
HOP62	Lung	G12C	30:10/11:01	30:10/03:01	07/56:01	44/40:01	05/01:02	07:02/03:04
LU65	lung	G12C	24:02	24:02	52:01	15:01	12:02	12:02
LU99	lung	G12C	24:02	24:02	54:01	52:01	01:02	01:02
MIA PACA-2	pancreas	G12C	24:02	24:02	14:02	14:02	08:02	08:02
NCI-H1373	lung	G12C	68:02	68:02	15:10	53:01	03:04	04:01
NCI-H1385	lung	G12C	03:01	34:02	58:02	58:02	06:02	06:02
NCI-H1792	lung	G12C	26:08	02:01	08:01	15:01	07:01	04:01
NCI-H2030	Lung	G12C	11:01	24:02	44:03	51:01	04:01	01:02
NCI-H2122	lung	G12C	03:01	01:01	08:01	07:02	07:01	07:01
NCI-H23	lung	G12C	80:01	80:01	50:01	50:01	06:02	06:02
SW 1573	lung	G12C	02:01	24:02	44:02	44:02	05:01	05:01
SW837	colon	G12C	32:01	32:01	35:01	35:01	04:01	04:01
NCI-H358	lung	G12C	03:01	03:01	35:01	15:01	04:01	03:04
SW1463	colon	G12C	01:01	02:01	08:01	44:02	07:06	05:01
AsPC-1	pancreas	G12D	01:01	26:01	15:01	15:01	03:04	03:04
CL40	colon	G12D	11:01	11:01	27:05	27:05	01:02	01:02
CSHL-0089	Organoid	G12D	32:01	32:01	39:01	35:01	04:01	12:03
CSHL-0090	Organoid	G12D	24:02	01:01	51:01	07:02	07:02	12:03
GP2d	colon	G12D	24:02	24:02	44:02	44:02	05:01	05:01
HPAF-II	pancreas	G12D	01:01	26:01	08:01	55:01	07:01	03:03
HuCC1	Biliary	G12D	11:01	11:01	51:01	44:02	15:02	08:02
LS 180	large_intestine	G12D	30:01	30:01	15:50	07:12	04:01	06:02
LS513	large_intestine	G12D	NA	NA	NA	NA	NA	NA
Panc 02.03	pancreas	G12D	02:05	31:01	49:01	08:01	07:01	07:01
Panc 04.03	pancreas	G12D	32:01	32:01	27:05	27:05	01:02	01:02
Panc 05.04	pancreas	G12D	11:01	02:06	55:01	27:05	03:03	03:03
Panc 08.13	pancreas	G12D	02:01	02:01	15:40	07:02	05:01	02:02
Panc 10.05	pancreas	G12D	01:01	31:01	40:01	49:01	07:01	03:04
PANC-1	pancreas	G12D	11:01	02:01	38:01	38:01	12:03	12:03
PK-1	pancreas	G12D	NA	NA	NA	NA	NA	NA
PK-45H	pancreas	G12D	33:03	33:03	44:03	44:03	14:03	14:03
PK-59	pancreas	G12D	33:03	31:01	58:01	35:01	04:01	03:02
SK-LU-1	lung	G12D	24:02	24:02	40:02	40:02	02:02	02:02
SNU-407	large_intestine	G12D	NA	NA	NA	NA	NA	NA
SNU-601	gastric	G12D	11:01	26:01	40:01	15:01	04:01	15:02
SNU-C2A	colon	G12D	24:02	26:01	48:01	07:02	14:02	08:03
SU.86.86	pancreas	G12D	NA	NA	NA	NA	NA	NA
SW 1990	pancreas	G12D	26:01	26:01	27:05	38:01	01:02	01:02
T3M-10	lung	G12D	24:02	11:01	46:01	40:01	01:03	03:10
CAL-62	thyroid	G12R	69:01	69:01	35:02	44:03	04:01	16:01
HuP-T3	pancreas	G12R	24:02	02:07	40:06	40:06	08:01	01:02
KP-2	pancreas	G12R	24:02	26:03	52:01	15:01	12:02	03:03
MDA-MB-134-V1	breast	G12R	24:02	11:01	40:01	35:01	03:04	04:01
PSN-1	pancreas	G12R	24:02	24:02	52:01	52:01	12:02	12:02
Capan-1	pancreas	G12V	30:01	01:01	13:02	57:01	06:02	06:02
CFPAC-1	pancreas	G12V	03:01	02:01	73:01	35:08	04:01	15:05
COLO 668	lung	G12V	NA	NA	NA	NA	NA	NA
COR-L23	lung	G12V	01:01		08:01		07:01	
CSHL-0063	Organoid	G12V	02:05	02:05	45:01	07:05	07:02	16:01
CSHL-0092	Organoid	G12V	02:01	01:01	51:01	52:01	01:02	12:02
DAN-G	pancreas	G12V	02:01	02:01	07:02	13:02	07:02	06:02
HUP-T4	pancreas	G12V	02:06	02:06	51:01	15:01	15:02	07:02
KP-3	pancreas	G12V	24:02	24:02	35:01	35:01	08:01	08:01
LCLC-97TM1	lung	G12V	02:19	24:02	15:01	18:01	03:03	12:03
NCI-H2444	lung	G12V	03:01	03:01	07:02	07:02	07:02	07:02
NCI-H441	lung	G12V	03:01	02:01	44:03	38:01	16:02	16:02
NCI-H727	lung	G12V	32:01	3:01	67:02	38:01	12:03	12:03
PA-TU-8902	pancreas	G12V	02:01	02:01	51:01	51:01	15:02	15:02
Panc 03.27	pancreas	G12V	24:02	02:01	07:02	56:01	01:02	07:02
QGP-1	pancreas	G12V	24:02	24:02	67:02	15:11	01:21	12:03
RCM-1	large_intestine	G12V	24:02	24:02	51:01	54:01	14:02	01:02
RERF-LC-Ad2	lung	G12V	02:01	24:02	13:01	48:01	03:04	04:01
SH-P-77	lung	G12V	24:02	02:01	40:08	54:01	03:08	07:01
SK-CO-1	colon	G12V	01:01	02:01	35:08	58:05	05:01	07:01
SNGM	endometrial	G12V	11:01	26:03	54:01	51:01	01:02	03:04
SW 900	lung	G12V	33:01	33:01	14:02	14:02	08:02	08:02
SW403	large_intestine	G12V	03:01	02:05	07:02	49:01	07:02	07:02
SW480	large_intestine	G12V	24:02	02:01	07:02	15:18	07:04	07:04
SW620	colon	G12V	24:02	02:01	07:13	37:04	07:04	07:04
SW626	large_intestine	G12V	NA	NA	NA	NA	NA	NA
YAPC	pancreas	G12V	24:02	24:02	52:01	35:01	12:02	12:02

REVIEWER COMMENTS

Reviewer #1 (Remarks to the Author):

The revised version of the manuscript of Bear et al addressed the main issues I was raising during my initial reviewing of the study.

The mouse experiments are an important improvement for the study and represent the circle closure point.

I have some further comments, which I think can help to improve the manuscript and its values for readers. In particular:

- p-HLA-I IC50.

As I was previously mentioning, the log values are kind of misleading. In Table S1 and S3 the predicted IC50 are reported as nM and as log nM. It shall be clarified somewhere that it is a log in base 10.

I suggest that in both Tables the authors write also the measured IC50 values in nM rather than only in log. Once they do that, it will be easy for the readers to see that the predicted IC50 values and the measured IC50 values are very different in terms of absolute values.

Let's take the 5-14V peptide; predicted IC50 for A0201 is 323 nM. Authors' measured IC50 is 2,944 nM. Measured IC50 for A0201 by Mishto et al., *Front. Immunol.* 2019 is around 170 nM (Sette lab assay).

Let's take the 8-16V peptide; predicted IC50 for A1101 is 54 nM. Authors' measured IC50 is 259 nM. Measured IC50 for A1101 (unpublished) is 7 nM (Sette lab assay).

By comparing these values, I would say that the Authors' assay is a kind of reliable relative quantification, although the absolute values are not those measured by others (Sette lab, in these cases).

I think the authors should briefly comment on it, report the Authors' measured IC50 as nM in Table S1 and S3, and also add an extra column with the IC50 measured by others, to include the present study in the framework of already published studies. For example, I estimate that the control peptides reported in Table S3 have a measured and published already IC50 value.

- MS data shown in Fig. S5-S7.

Through a second look at the MS2 shown in Fig. S5-S7, I have some doubts about some of the identified peptides such as Fig. S6a and S6e. Did the Authors compute the doc product considering all peaks or only the matched peaks? I am surprised that the idopt for the peptide in Fig. S6e is 0.96. The authors should: (a) change the representation of the ions so that the not assigned and assigned ions have the same stroke size and not assigned ions may be in black so that readers can distinguish them; (b) all MS RAW files must be available on online repositories such as PRIDE. This is a standard practise for proteomics, which will allow validation (and use) of the RAW files by others. I checked the files in the PRIDE dataset and I found only 2 RAW files, no description of the files. I reckon they are the Fusion Lumos files. I expected to find all the RAW files of the QE HF, which led to the identification of the mKRAS in the immunopeptidomes. Did I miss anything? Are they deposited somewhere else?

- Reference to literature.

As previously mentioned in my comments, the study focused on mutations and epitopes that were in part described already. The study is now complete enough to compensate this partial lack of novelty. However, a precise reference to previous papers would be helpful.

For example, I appreciate that the Authors refer also to non-canonical (spliced) KRAS neoepitopes in their discussion. However, Mishto et al., *Front Immunol.* 2019 did not study the HLA-I immunopeptidomes of mKRAS cell lines. They did, by contrast, prove that a spliced neoepitope AND the 5-14V peptide were produced by proteasomes. The latter info is in contract to what written by the Authors in the revised Discussion.

Furthermore, the peptide 7-16V has been proved to be produced by proteasomes in in vitro digestions, as published by Specht et al., *Sci. Data* 2020.

The Authors may consider to rephrase their discussion and report these info, as well as others on the mKRAS epitope they described, to strengthen the manuscript and to provide a full picture to the readers.

- Data availability.

I think that the statement "The data that supports the findings of this study are available from the corresponding author upon reasonable request." is not what we expect in times of full sharing of data within the scientific community. All data needed to reproduce the study, and to use the data to move forward the research, should be available online and well described, rather than leaving to the corresponding author the power to decide what a REASONABLE request is.

For example, see my comment about the PRIDE dataset. All proteomics files should be uploaded and well described

The same concept is applicable for the TCR alpha sequence of the TCRs described in the manuscript. It would be extremely useful for the community if the TCRs sequence were disclosed. If they have been patented, there should be no problem, no?

Reviewer #2 (Remarks to the Author):

The authors have thoroughly addressed all of my prior concerns, which were in large part related to issues of data presentation. My original high level of enthusiasm remains for this important work.

Reviewer #3 (Remarks to the Author):

I find this to be an excellent manuscript and I support publication.

Reviewer #4 (Remarks to the Author):

The efforts made by the authors are highly appreciated and the manuscript has gained substantially by the revision. However, my main concerns about the specificity of the described TCR have unfortunately not been fully addressed and were answered unsatisfactory by the comments made by the authors:

(1) Usage of mutated and wildtype minigenes for validation of neoantigen specificity of respective TCR

Answer by the authors: The suggestion to introduce a KRAS mutation in WT cells to create isogenic cell line is appreciated and we will certainly consider this approach in the future.

Response: Transduction of mutated and wildtype tandem mini-gene constructs (TMC) in antigen negative HLA-matched cells has been previously published many times as appropriate state of the art control tool to demonstrate (neo)antigen specificity (1, 2). Peptide pulsing of K562 cells is not sufficient as this may block/reduce recognition of alternative target antigens detected by these TCR.

(2) Investigation of the cross-reactivity potential of the respective TCR by alternative peptide scans in order to determine the specificity of the TCR and excluding major cross-reactivity potentially involved in recognition of targets used in the experiments.

Answer by the authors: Thank you for the suggestion but we should have been clearer regarding the primary objective of our study which was to focus on epitope discovery and validation. We employed each TCR as a reagent to probe and detect mKRAS peptide/HLA complexes on human tumor cells. Assessment of individual TCR attributes such as potential cross-reactivity to non-cognate peptides is beyond the scope of our report. As we isolate and characterize additional TCRs, we plan to investigate this issue as suggested but for now, this exercise will require significant resources and, in our opinion, will not alter the conclusions of our study.

Response: The objective of the study is demonstrating specific immune responses against mutated

KRAS. As investigation of neoantigen specificity has not been performed by using TMC-transduced antigen-negative cells, such cross-reactivity assays would give additional information about the cross-reactivity potential of defined TCR and the probability that observed reactivity may be independent of the presentation of mutated KRAS peptides. I wonder why such simple cross-reactivity analyses as alanine/threonine scans were not performed. It is not sufficient to test exclusively the G12 peptide panel.

In order to exclude recognition of alternative mKRAS-independent peptides as targets which are responsible for tumor eradication by described TCR, analyses including controls described above would need to be accomplished.

1. Robbins PF, Lu YC, El-Gamil M, Li YF, Gross C, Gartner J, et al. Mining exomic sequencing data to identify mutated antigens recognized by adoptively transferred tumor-reactive T cells. *Nat Med* 2013;19(6):747-52
2. Parkhurst MR, Robbins PF, Tran E, Prickett TD, Gartner JJ, Jia L, et al. Unique Neoantigens Arise from Somatic Mutations in Patients with Gastrointestinal Cancers. *Cancer Discov* 2019;9(8):1022-35

To Reviewers:

We thank each reviewer for their suggestions and hope that our response adequately addresses the points raised.

Reviewer 1

The revised version of the manuscript of Bear et al addressed the main issues I was raising during my initial reviewing of the study.

The mouse experiments are an important improvement for the study and represent the circle closure point.

Response: **Thank you for the positive comments.**

Comment 1 - p-HLA-I IC50.

As I was previously mentioning, the log values are kind of misleading. In Table S1 and S3 the predicted IC50 are reported as nM and as log nM. It shall be clarified somewhere that it is a log in base 10. I suggest that in both Tables the authors write also the measured IC50 values in nM rather than only in log. Once they do that, it will be easy for the readers to see that the predicted IC50 values and the measured IC50 values are very different in terms of absolute values. Let's take the 5-14V peptide; predicted IC50 for A0201 is 323 nM. Authors' measured IC50 is 2,944 nM. Measured IC50 for A0201 by Mishto et al., Front. Immunol. 2019 is around 170 nM (Sette lab assay). Let's take the 8-16V peptide; predicted IC50 for A1101 is 54 nM. Authors' measured IC50 is 259 nM. Measured IC50 for A1101 (unpublished) is 7 nM (Sette lab assay). By comparing these values, I would say that the Authors' assay is a kind of reliable relative quantification, although the absolute values are not those measured by others (Sette lab, in these cases). I think the authors should briefly comment on it, report the Authors' measured IC50 as nM in Table S1 and S3, and also add an extra column with the IC50 measured by others, to include the present study in the framework of already published studies. For example, I estimate that the control peptides reported in Table S3 have a measured and published already IC50 value.

Response: **We have now added the log base 10 reference throughout the manuscript and have included \log_{10} nM and nM values in Tables S1 and S3 as determined using competitive binding assays. These assays experimentally confirm the binding of peptides to their predicted HLA class I allele and we make no claims that these measurements represent absolute values. We have previously reported our melanoma neoantigen data (Table S4, Carreno et al., science.aaa3828; Table S4, Linette et al., pnas.1906026116) and these assays have been broadly accepted as experimental binding validation. As the reviewer may appreciate, we cannot comment on affinity values not reported in the literature and to our knowledge only Mishto et al. have reported data for the 5-14V peptide (mentioned in Discussion section, p. 17, line 406).**

Comment 2 -MS data shown in Fig. S5-S7.

Through a second look at the MS2 shown in Fig. S5-S7, I have some doubts about some of the identified peptides such as Fig. S6a and S6e. Did the Authors compute the doc product considering all peaks or only the matched peaks? I am surprised that the idopt for the peptide in Fig. S6e is 0.96. The authors should: (a) change the representation of the ions so that the not assigned and assigned ions have the same stroke size and not assigned ions may be in black so

that readers can distinguish them; (b) all MS RAW files must be available on online repositories such as PRIDE. This is a standard practise for proteomics, which will allow validation (and use) of the RAW files by others. I checked the files in the PRIDE dataset and I found only 2 RAW files, no description of the files. I reckon they are the Fusion Lumos files. I expected to find all the RAW files of the QE HF, which led to the identification of the mKRAS in the immunopeptidomes. Did I miss anything? Are they deposited somewhere else?

Response: Regarding the idotp and dotp comparison: As detailed by the Skyline software, only matching ions for the idotp and dotp are used for comparisons. Since all our targets were 2+ charged peptides, the matching ions in each comparison included the precursors: [M]2+, [M+1]2+, [M+2]2+ and the products: y-series 1+ and b-series 1+. This is standard in the analysis performed by Skyline and the reported values were derived accordingly.

The Proteomics methods section in the manuscript has been edited for clarification. A key (in excel format) to navigate the various files has been deposited along with MS Raw data, DDA and PRM files in PRIDE (PDX024412). Username: reviewer_pxd024412@ebi.ac.uk, password: Gk722zvc

Comment 3 - Reference to literature.

As previously mentioned in my comments, the study focused on mutations and epitopes that were in part described already. The study is now complete enough to compensate this partial lack of novelty. However, a precise reference to previous papers would be helpful.

For example, I appreciate that the Authors refer also to non-canonical (spliced) KRAS neoepitopes in their discussion. However, Mishto et al., Front Immunol. 2019 did not study the HLA-I immunopeptidomes of mKRAS cell lines. They did, by contrast, prove that a spliced neoepitope AND the 5-14V peptide were produced by proteasomes. The latter info is in contract to what written by the Authors in the revised Discussion.

Furthermore, the peptide 7-16V has been proved to be produced by proteasomes in in vitro digestions, as published by Specht et al., Sci. Data 2020. The Authors may consider to rephrase their discussion and report these info, as well as others on the mKRAS epitope they described, to strengthen the manuscript and to provide a full picture to the readers.

Response: Please refer to Discussion section (p. 17, Line 406) for modifications related to this point. We have corrected the error regarding in vitro proteasome generation of 5-14V.

Comment 4 - Data availability

I think that the statement "The data that supports the findings of this study are available from the corresponding author upon reasonable request." is not what we expect in times of full sharing of data within the scientific community. All data needed to reproduce the study, and to use the data to move forward the research, should be available online and well described, rather than leaving to the corresponding author the power to decide what a REASONABLE request is. For example, see my comment about the PRIDE dataset. All proteomics files should be uploaded and well described. The same concept is applicable for the TCR alfabeta sequence of the TCRs described in the manuscript. It would be extremely useful for the community if the TCRs sequence were disclosed. If they have been patented, there should be no problem, no?

Response: The following statement has been included in the manuscript (p. 20, Line 476): "MS Raw data, DDA and PRM files used in this study has been deposited in PRIDE repository with the database identifier PDX024412. All other data generated or analyzed during this study are

included in this published article and its Supplementary Data files. The source data underlying both the main and Supplementary Figs. are provided as a Source Data file.”

Regarding the data for TCR alpha / beta sequences, it is standard practice (in NCOMMS as well as other journals) to report the V-D-J gene segments as well as the CDR3 regions as AA sequences. This information (provided in the initial submission) may be found in Fig 2E.

Reviewer 4

The efforts made by the authors are highly appreciated and the manuscript has gained substantially by the revision.

Response: Thank you for the positive comments.

However, my main concerns about the specificity of the described TCR have unfortunately not been fully addressed and were answered unsatisfactory by the comments made by the authors:

(1) Usage of mutated and wildtype minigenes for validation of neoantigen specificity of respective TCR.

Answer by the authors: The suggestion to introduce a KRAS mutation in WT cells to create isogenic cell line is appreciated and we will certainly consider this approach in the future.

Response: Transduction of mutated and wildtype tandem mini-gene constructs (TMC) in antigen negative HLA-matched cells has been previously published many times as appropriate state of the art control tool to demonstrate (neo)antigen specificity (1, 2). Peptide pulsing of K562 cells is not sufficient as this may block/reduce recognition of alternative target antigens detected by these TCR.

Response to Reviewer: The use of TMC, by Rosenberg’s group, we (Carreno et al., science.aaa3828) and others is one of many tools to evaluate antigen processing and presentation. The TMC method was developed by our group to evaluate antigen processing via the proteasome and **not** to define antigen specificity. In the 2 papers referenced by the reviewer, both TMC **and** (minimal epitope) peptide pulsing experiments are performed (for example Fig 1a, d & g in Robbins et al. [ref#1] and SFig. 6 in Parkhurst et al. [ref#2]) to fully characterize antigen recognition. Antigen specificity is best evaluated by studying minimal/optimal epitope bound to MHC class I as we have shown in our manuscript. However, to further address the reviewer’s point, we have now included new data using WT KRAS-expressing cell (BxPC-3): (1) as targets in functional assays using mKRAS TCR-engineered T cells (Fig 4) and (2) as APC in reporter assays using mKRAS TCRA3V, TCRA11V and TCRB7R-J^{ASP90} cells (SFig 10). Both of these experiments demonstrate that TCRs described in our study fail to recognize WT KRAS as well as other alternative peptides that may be presented in the absence of cognate antigen. Finally, the notion that peptide pulsing in K562/HLA class I-expressing transfectants may block/reduce recognition of alternative target antigens is speculative and highly improbable for 3 unique TCRs based on the fact that no reactivity is seen for control (no peptide) K562/HLA class I-expressing transfectants.

(2) Investigation of the cross-reactivity potential of the respective TCR by alternative peptide scans in order to determine the specificity of the TCR and excluding major cross-reactivity potentially involved in recognition of targets used in the experiments.

Answer by the authors: Thank you for the suggestion but we should have been clearer regarding the primary objective of our study which was to focus on epitope discovery and validation. We employed each TCR as a reagent to probe and detect mKRAS peptide/HLA complexes on human tumor cells. Assessment of individual TCR attributes such as potential cross-reactivity to non-cognate peptides is beyond the scope of our report. As we isolate and characterize additional TCRs, we plan to investigate this issue as suggested but for now, this exercise will require significant resources and, in our opinion, will not alter the conclusions of our study.

Response: The objective of the study is demonstrating specific immune responses against mutated KRAS. As investigation of neoantigen specificity has not been performed by using TMC-transduced antigen-negative cells, such cross-reactivity assays would give additional information about the cross-reactivity potential of defined TCR and the probability that observed reactivity may be independent of the presentation of mutated KRAS peptides. I wonder why such simple cross-reactivity analyses as alanine/threonine scans were not performed. It is not sufficient to test exclusively the G12 peptide panel.

In order to exclude recognition of alternative mKRAS-independent peptides as targets which are responsible for tumor eradication by described TCR, analyses including controls described above would need to be accomplished.

Response to Reviewer: As previously stated, the goal of our study is to identify and characterize mKRAS as candidate MHC I neoantigens. We have now included results with WT KRAS expressing cell lines (Fig 4 and SFig 10) and hope this data address the reviewer's concerns regarding TCR specificity. We stand by our previous comments that evaluation of TCR cross-reactivity using alanine scanning is outside the objective of the current study.

The request to "exclude recognition of alternative mKRAS-independent peptides as targets which are responsible for tumor eradication by described TCRs" is illogical based on first principles. Both mKRAS TCRs fail to react to COR-L23 in the absence of HLA-A*03:01/A*11:01 (despite the expression of endogenous A*01:01/B*08:01/C*07:01) in various in vitro assays. There is no hint or suggestion for recognition of unrelated self-peptides using the panel of tumor cell lines shown in STable 4. The mKRAS target peptides are quantitated and reproducibly found on the surface of COR-L23/A*03:01 and COR-L23/A*11:01 lung cancer cells. Thus we stand firm in our position, one supported by Fig 2 in reference #2 quoted by reviewer, that the assays included in our study are sufficient for defining TCR antigen specificity.

1. Robbins PF, Lu YC, El-Gamil M, Li YF, Gross C, Gartner J, et al. Mining exomic sequencing data to identify mutated antigens recognized by adoptively transferred tumor-reactive T cells. *Nat Med* 2013;19(6):747-52
2. Parkhurst MR, Robbins PF, Tran E, Prickett TD, Gartner JJ, Jia L, et al. Unique Neoantigens Arise from Somatic Mutations in Patients with Gastrointestinal Cancers. *Cancer Discov* 2019;9(8):1022-35

The End

REVIEWERS' COMMENTS

Reviewer #1 (Remarks to the Author):

The authors responded to all my questions.

Reviewer #4 (Remarks to the Author):

The efforts made by the authors are appreciated. I think adding the additional control in Figure 4 and S10 is helpful to exclude major cross reactivity of defined TCR in this cell line although this cannot be excluded to be the case in other cell lines used. The reason why TMC have not not been applied is not reasonably explained as this is a wonderful control to show specificity even if this was not the intention by the authors when using this control first.

To Reviewers:

Reviewer 1

The authors responded to all my questions.

Response: **Thanks!**

Reviewer 4

The efforts made by the authors are appreciated. I think adding the additional control in Figure 4 and S10 is helpful to exclude major cross reactivity of defined TCR in this cell line although this cannot be excluded to be the case in other cell lines used. The reason why TMC have not not been applied is not reasonably explained as this is a wonderful control to show specificity even if this was not the intention by the authors when using this control first.

Response: **Thank you for the positive comments. We have now included in the Results section a description detailing the use of exogenous mutated and wild-type KRAS peptides to examine antigen specificity and TMC to evaluate mutated KRAS epitope processing and presentation.**

The End